# DPaI: Differentiable Pruning at Initialization with Node-Path Balance Principle

**Lichuan Xiang[1,4,*], Quan Nguyen-Tri[2,3,*], Lan-Cuong Nguyen[2,3], Hoang Pham[1],**

**Khoat Than[2], Long Tran-Thanh[1], Hongkai Wen[1,†]**
[1]University of Warwick, [2]Hanoi University of Science and Technology,
[3]FPT Software AI Center, [4]Collov Labs

## Abstract

Pruning at Initialization (PaI) is a technique in neural network optimization characterized by the proactive elimination of weights before the network's training on designated tasks. This innovative strategy potentially reduces the costs for training and inference, significantly advancing computational efficiency. A key factor leading to PaI's effectiveness is that it considers the saliency of weights in an untrained network, and prioritizes the trainability and optimization potential of the pruned subnetworks. Recent methods can effectively prevent the formation of hard-to-optimize networks, e.g. through iterative adjustments at each network layer. However, this way often results in *large-scale discrete optimization problems*, which could make PaI further challenging. This paper introduces a novel method, called *DPaI*, that involves a differentiable optimization of the pruning mask. DPaI adopts a dynamic and adaptable pruning process, allowing easier optimization processes and better solutions. More importantly, our differentiable formulation enables readily use of the existing rich body of efficient gradient-based methods for PaI. Our empirical results demonstrate that DPaI significantly outperforms current state-of-the-art PaI methods on various architectures, such as Convolutional Neural Networks and Vision-Transformers. Code is available at https://github.com/QuanNguyen-Tri/DPaI.git

## 1 Introduction

The Lottery Ticket Hypothesis (LTH) (Frankle & Carbin, 2018; Chen et al., 2020; 2021a) serves as a foundational concept in our research, revealing the potential of sparse subnetworks that can be trained from scratch to achieve the performance levels of their dense counterparts. However, a critical challenge with LTH is its resource-intensive nature, involving an iterative process of pruning and retraining that often exceeds the cost of training a dense network. This challenge presents an intriguing research question: Can we identify sparse, trainable subnetworks at the initialization phase, eliminating the need for pre-training? Specifically, a method capable of effective pruning before training could significantly reduce memory and computational costs without substantially impacting performance (Wang et al., 2022). Such a breakthrough would increase the adaptability of neural networks in resource-constrained environments (Alizadeh et al., 2022; Yuan et al., 2021a).

To address this, various Pruning at Initialization (PaI) methods have been proposed (Lee et al., 2019b; Tanaka et al., 2020; de Jorge et al., 2021; Wang et al., 2020; Alizadeh et al., 2022; Liu et al., 2022b). These methods often rely on gradient information (Lee et al., 2019b; Wang et al., 2020) and focus on assessing the importance of network parameters. Nonetheless, their effectiveness in reducing computational load and maintaining accuracy is limited. This limitation arises from a singular focus on parameter importance, neglecting the broader network topology, leading to sub-optimal, hard-to-optimize narrow networks during retraining.

Recent studies (Frankle et al., 2021; Su et al., 2020) suggest that the overall topology of the final sparse network may be more crucial than individual weight-wise importance. Building on these

---

*Equal Contribution.
†Corresponding author.

findings, (Pham et al., 2023) have introduced a Node-Path Balancing Principle (NPB) that shows the balance between the number of effective nodes and the effective path is essential for generating extremely sparse networks with good performance. However, to do so, the current NPB principle requires addressing the complexity of an underlying sequence of discrete optimization problems, which necessitates iterative solutions at each layer, often resulting in sub-optimal outcomes for the final pruning mask.

In this paper, we introduce a novel approach that resolves the mask optimization issue at initialization in a learnable manner, directly targeting the overall network metrics and enhancing performance. In particular, our approach makes NPB differentiable (and thus, making the NPB principle more compatible with standard neural network training processes, a.k.a. learnable) by replacing the current discrete (i.e., non-differentiable) optimization component in the NPB principle with a new differentiable module. A *key challenge here is to relax the underlying set of non-linear integer programs into a continuous version, which still provides valid solutions*. We introduce DPaI (**D**ifferentiable **P**runing **at I**nitialization), a novel differentiable continuous optimization approach to identify good final pruning masks. In summary, our contributions are as follows:

- DPaI is the first *differentiable* pruning at initialization method that takes into account network topology, specifically the Node-path Balancing Principle (NBP), a recently identified essential concept to achieve good sparse neural networks.
- It extends NPB into a differentiable formulation, making its integration with the training process of sparse neural networks more natural. Different from NPB, our DPaI enables readily use of the existing rich body of efficient gradient-based methods.
- Extensive experiments on diverse datasets show that the proposed DPaI can discover superior sparse sub-networks across multiple sparsity levels.

## 2 RELATED WORK

**Neural Network Pruning.** Traditional neural network pruning methods, as described in seminal works by LeCun et al. (1989), Hassibi et al. (1993), Han et al. (2015), Dong et al. (2017), and Molchanov et al. (2017), primarily focus on trimming trained models based on specific pre-defined criteria. Subsequently, these pruned models, or subnetworks, undergo fine-tuning to ensure convergence. However, recent empirical studies by Frankle & Carbin (2018), Frankle et al. (2020), Chen et al. (2020), and Chen et al. (2021a) have revealed the existence of 'lottery tickets' - subnetworks with random initializations that, when trained from scratch or in the early stages of training, can perform comparably to their unpruned, dense counterparts. Despite these findings, the process of identifying these 'lottery tickets' remains computationally intensive due to the necessity of repeated training and pruning cycles, as highlighted by Frankle et al. (2020) and Frankle & Carbin (2018).

In response to this challenge, gradual pruning approaches such as those proposed by Zhu & Gupta (2017) and Gale et al. (2019), intertwine the processes of pruning and training. These methods typically incur lower computational costs compared to post-training pruning. Nonetheless, they still require initial training phases to determine the most effective sparse subnetwork configuration. In contrast, other techniques, such as those introduced by Chen et al. (2021b) and Chen et al. (2023), implement one-shot pruning during the training process to further decrease computational demands. Alternatively, dynamic sparse training methodologies, as explored by Mocanu et al. (2018), Evci et al. (2020), Liu et al. (2021b), and Liu et al. (2021a), initiate with a randomly sparse network and adaptively update network connectivity throughout the training.

Moreover, certain approaches (e.g. Lee et al. (2019b), Wang et al. (2020), Patil & Dovrolis (2021), Tanaka et al. (2020), and Alizadeh et al. (2022)) focus on determining subnetworks based on network initialization, gradient information, and network topology, essentially pruning before training. However, as Frankle et al. (2021) and Su et al. (2020) have demonstrated through their experimental results, the existing criteria in these PaI methods may not always yield subnetworks with optimal performance.

**Network Shape and Gradients During Pruning.** In Pruning at initialization (PaI) methods, where training data is often unused (Tanaka et al., 2020; Patil & Dovrolis, 2021) or used minimally for gradient information (Lee et al., 2019b; Wang et al., 2020), network topology becomes critical to optimize pruned network performance. Various PaI techniques affect network topology. SynFlow (Tanaka et al., 2020) aims to maintain input-output paths but often increases isolated neurons.

Strategies by Patil & Dovrolis (2021) and Gebhart et al. (2021) preserve network efficiency through path kernels, focusing on subnetwork structure. PHEW (Patil & Dovrolis, 2021) uses random walks to increase effective nodes but reduces input-output paths. Other researchers have examined 'effective nodes' and 'effective paths' in pruned subnetworks (Wang et al., 2020; Naji et al., 2021).

Recent advancements in Neural Architecture Search (NAS) by Xiang et al. (2023) and Sun et al. (2023) integrate gradient signal-to-noise ratio in zero-cost network performance evaluations. This method balances gradient distribution across layers, ensuring uniform spread and avoiding gradient concentration, which can lead to narrow layers and poor performance. This approach aligns with node-path balance, considering the quality of gradients along various paths for a more effective evaluation of network trainability.

**Extreme Sparse Networks.** In the context of extreme sparsity (Cho et al., 2021; Price & Tanner, 2021), the network density is less than $1\%$. Cho et al. (2021) associate two works from Lee et al. (2019b); Zhou et al. (2019) to learn masking during training. Tanaka et al. (2020); de Jorge et al. (2021); Vysogorets & Kempe (2021) leverage iterative pruning to prevent subnetworks from layer collapse in super sparsity cases. Price & Tanner (2021) only requires an extremely small number of trainable parameters associated with a freeze fully connected network, which helps the model performing well on extreme sparsity settings. This suggests preserving information flow through network connections is crucial in extremely sparse networks.

**Differentiable Neural Architecture Search.** Liu et al. (2019) introduced the concept of using continuous architectural parameters ($\alpha$) for searching neural network architectures in a differentiable manner, optimizing $\alpha$ with $\nabla_\alpha \mathcal{L}_v(w - \xi \nabla_w \mathcal{L}_t(w, \alpha))$. This method involves constructing a *supernet* that encompasses all architectures within the search space and optimizing both $\alpha$ and the supernet weights ($w$). The final architecture is derived by retaining operations with the highest $\alpha$ values. Despite reducing search time, DARTS has stability and generalizability issues, often favouring trivial models with excessive skip connections (Zela et al., 2020). To address these issues, SDARTS (Chen & Hsieh, 2020) smooths the loss landscape, and SGAS (Li et al., 2020) uses a greedy algorithm for operational selection and pruning. Recently, DARTS-PT (Wang et al., 2021) proposed a perturbation-based operation selection method, evaluating operations based on their impact on the supernet's validation accuracy upon removal. However, while these approaches derive dense subnetworks from the supernet by learning and optimizing $\alpha$ during training, they may not be suited for identifying sparse subnetworks at initialization.

## 3 METHOD

In the following section, we first describe the NPB principle (Section 3.1), which forms the basis of our solution. We discuss the novel formulation of differentiable Node-Path Balancing (d-NPB) in Section 3.1, and then present our DPaI algorithm in Section 3.3, followed by convergence analysis for d-NPB methods in Section 3.4.

### 3.1 THE NODE-PATH BALANCING PRINCIPLE

In a sparse network, it is intuitively clear that one should arrange the connections into a configuration that is neither too thin nor too spread out to have good information propagation during training. Evidence of this understanding is the Node-Path Balancing (NPB) principle introduced by Pham et al. (2023). In what follows, we first start with the key definitions, then introduce the NPB concept; the detail of metrics implementation can be found in Appendix I:

**Effective Path.** We define a path to be *effective* if it connects an input node to an output node without interruption. Metrics based on paths are mentioned in Tanaka et al. (2020); Gebhart et al. (2021) as $l_1$ and $l_2$ path norms, respectively. In this paper, we only consider the number of paths.

**Effective Node/Channel.** A node/channel is effective if at least one effective path goes through it. This concept is also considered in Patil & Dovrolis (2021); Frankle et al. (2021). For convolutional layers, we consider a kernel as a connection and a channel as a node and then convert the convolutional layer into a fully connected layer.

The key objective of the NPB principle is to generate a sparse network with the maximal number of effective nodes and paths simultaneously, given the sparsity constraint. To achieve this, the principle aims to solve the following discrete optimization problem:

Given an architecture $A$ with parameter $\mathbf{W} \in \mathbb{R}^N$, where $N$ is the total number of parameters, and sparsity ratio $\tau$. Denote $\mathcal{R}_P$ as the total number of input-output paths, $\mathcal{R}_N$ as the number of activated nodes, and consider the mask for parameter $\mathbf{M} = \{0, 1\}^N$ as the variable to solve. For some $0 \leq \alpha \leq 1$:

$$\text{Maximize} \quad \mathcal{R}_{NPB} := \alpha \mathcal{R}_N + (1 - \alpha)\mathcal{R}_P \quad \text{s.t.} \quad \|\mathbf{M}\|_1 \leq N(1 - \tau)$$

This large-scale discrete optimization problem is NP-hard, and thus, Pham et al. (2023) have proposed a heuristic to produce sparse neural network architectures. However, this heuristic is often suboptimal, and the underlying discrete optimization concept cannot be easily integrated into the standard neural network training pipeline, which may limit the usage of the NPB principle.

### 3.2 DIFFERENTIABLE NODE-PATH BALANCING (D-NPB) OPTIMIZATION

To overcome this issue, this section introduces a novel way of converting the discrete NPB principle into a differentiable form. To do so, we introduce a differentiable parameter $s^{(l)} \in \mathbb{R}^{h^{(l-1)} \times h^{(l)}}$ to adjust the importance of parameters in layer $l$, in which $h^{(l)}$ is the number of neurons in layer $l$. The mask for the parameters in layer $l$ can be obtained by: $m_{i,j}^{(l)} = \text{Top}_{k^{(l)}}(|s_{i,j}^{(l)}|)$, where $\text{Top}_k(x) = \begin{cases} 1 & \text{if } x \text{ in } k \text{ largest elements} \\ 0 & \text{otherwise} \end{cases}$, and we set $m_{i,j}^{(l)} = \begin{cases} 0 & \text{if pruned} \\ 1 & \text{otherwise} \end{cases}$, $k^{(l)}$ can be the desired density level for layer $l$ (e.g. given by the Erdős-Rényi Kernel (ERK) (Liu et al., 2022a) ).

Due to the extremely large number of paths, especially in denser networks, there is an imbalance with other objectives, like the number of nodes, which is significantly smaller. To address this, we applied a logarithmic scale to all objectives to prevent this imbalance. The optimisation of the NPB principle now becomes:

$$\text{Maximize} \quad \mathcal{R}_{NPB} := \alpha \log \mathcal{R}_N + (1 - \alpha) \log \mathcal{R}_P \quad \text{s.t.} \quad \|\mathbf{M}\|_1 \leq N(1 - \tau)$$

**Differentiable Calculation of the Effective Path**: Denotes $P(v_j^{(l)})$ is the number of incoming paths to a node $v_j^{(l)}$. The number of effective paths is the number of incoming paths to the nodes in the last layer $L$:

$$\mathcal{R}_P = \sum_{j=1}^{h^{(L)}} P(v_j^{(L)}), \quad P(v_j^{(l)}) = \sum_{i=1}^{h^{(l-1)}} m_{i,j}^{(l)} P(v_i^{(l-1)}), \quad P(v_j^{(0)}) = 1 \tag{1}$$

The derivative of $s_{i,j}^{(l)}$ can be estimated via the Straight-Through Estimator (Bengio et al., 2013) (the derivative goes "straight-through" $\text{Top}_k(.)$):

$$\frac{\delta \log \mathcal{R}_P}{\delta s_{i,j}^{(l)}} = \frac{\delta \log \mathcal{R}_P}{\delta \mathcal{R}_P} \frac{\delta \mathcal{R}_P}{\delta P(v_j^{(l)})} \frac{\delta P(v_j^{(l)})}{\delta s_{i,j}^{(l)}} = \frac{1}{\mathcal{R}_P} \frac{\delta \mathcal{R}_P}{\delta P(v_j^{(l)})} P(v_i^{(l-1)}) \frac{|s_{i,j}^{(l)}|}{s_{i,j}^{(l)}} \tag{2}$$

The number of outgoing paths from node $v_j^{(l)}$ to the nodes in the last layer $L$ can be estimated by the following derivative:

$$\begin{aligned}
\frac{\delta \mathcal{R}_P}{\delta P(v_j^{(l)})} &= \sum_k \frac{\delta \mathcal{R}_P}{\delta P(v_k^{(l+1)})} \frac{\delta P(v_k^{(l+1)})}{\delta P(v_j^{(l)})} = \sum_k \frac{\delta \mathcal{R}_P}{\delta P(v_k^{(l+1)})} m_{j,k}^{(l+1)} \\
&= \sum_{n,p,q,\dots,k} \frac{\delta \mathcal{R}_P}{\delta P(v_n^{(L)})} m_{p,n}^{(L)} m_{q,p}^{(L-1)} \dots m_{j,k}^{(l+1)} = \sum_{n,p,q,\dots,k} m_{p,n}^{(L)} m_{q,p}^{(L-1)} \dots m_{j,k}^{(l+1)}
\end{aligned} \tag{3}$$

**Differentiable Calculation of the Effective Node/Channel**: A node $v_j^{(l)}$ is activated if there is a path connecting the input nodes to it $P(v_j^{(l)}) > 0$, and there is a path connecting it to the output nodes $\frac{\delta \mathcal{R}_P}{\delta P(v_j^{(l)})} > 0$. Therefore, a node $v_j^{(l)}$ is considered effective when :

$$N(v_j^{(l)}) = P(v_j^{(l)}) \frac{\delta \mathcal{R}_P}{\delta P(v_j^{(l)})} = \frac{\delta \mathcal{R}_P}{\delta P(v_j^{(l)})} \sum_{i=1}^{h^{(l-1)}} m_{i,j}^{(l)} P(v_i^{(l-1)}) \propto \sum_{i=1}^{h^{(l-1)}} \left| \frac{\delta \log \mathcal{R}_P}{\delta s_{i,j}^{(l)}} \right| m_{i,j}^{(l)} > 0 \tag{4}$$

We use $\tanh(x) = \frac{e^x - e^{-x}}{e^x + e^{-x}}$ as a differentiable activation function for the number of effective nodes counts, resulting in $\tanh\left(\gamma N(v_j^{(l)})\right) = 1$ when $v_j^{(l)}$ is an effective node, and $\gamma$ is a sufficiently large constant. The objective of maximizing the number of effective nodes can be written as follows:

$$\mathcal{R}_N = \sum_{l=1}^{L} \sum_{j=1}^{h^{(l)}} \tanh\left(\gamma N(v_j^{(l)})\right) \tag{5}$$

We can write the derivative of $\log \mathcal{R}_N$ as follows:

$$\frac{\delta \log \mathcal{R}_N}{\delta s_{i,j}^{(l)}} = \frac{\gamma}{\mathcal{R}_N} \left(1 - \tanh^2\left(\gamma N(v_j^{(l)})\right)\right) \frac{\delta \log \mathcal{R}_P}{\delta s_{i,j}^{(l)}} \tag{6}$$

**Differentiable Calculation of the Effective Kernel/Connection**: In a pruned network, we want to maximize the number of weights that receive gradient for the update. Pham et al. (2023) also integrates this idea in terms of a regularization that aims to encourage activating as many kernels as possible. We transfer this concept into our differentiable method via the concept of Effective Kernel/Weight. A kernel/connection $m_{i,j}^{(l)}$ is effective if there is an effective path pass through it: $N(m_{i,j}^{(l)}) = P(v_i^{(l-1)}) m_{i,j}^{(l)} \frac{\delta \mathcal{R}_P}{\delta P(v_j^{(l)})} \propto \left|\frac{\delta \log \mathcal{R}_P}{\delta s_{i,j}^{(l)}}\right| m_{i,j}^{(l)} > 0$. The objective for kernel/connection is: $\mathcal{R}_C = \sum_{l=1}^{L} \sum_{j}^{h^{(l)}} \sum_{i}^{h^{(l-1)}} \tanh\left(\gamma N(m_{i,j}^{(l)})\right)$. The derivative of $\log \mathcal{R}_C$ is computed as follows: $\frac{\delta \log \mathcal{R}_C}{\delta s_{i,j}^{(l)}} = \frac{\gamma}{\mathcal{R}_C}\left(1 - \tanh^2\left(N(m_{i,j}^{(l)})\right)\right)\frac{\delta \log \mathcal{R}_P}{\delta s_{i,j}^{(l)}}$. Note that $m_{i,j}^{(l)}$ represents a single connection in a fully connected layer or the whole kernel in a convolutional layer.

Then, we maximize the overall objective $\mathcal{R}_{DPaI} = (1-\alpha) \log \mathcal{R}_P + \alpha[(1-\beta) \log \mathcal{R}_N + \beta \log \mathcal{R}_C]$ by repeatedly computing the following rule until convergence, where $\eta$ is the learning rate:

$$s_{i,j}^{(l)} := s_{i,j}^{(l)} + \eta \left((1-\alpha)\frac{\delta \mathcal{R}_P}{\delta s_{i,j}^{(l)}} + \alpha \left((1-\beta)\frac{\delta \mathcal{R}_N}{\delta s_{i,j}^{(l)}} + \beta \frac{\delta \mathcal{R}_C}{\delta s_{i,j}^{(l)}}\right)\right) \tag{7}$$

### 3.3 Convergence analysis of Differentiable Node-Path Balancing

Assuming that after an update, edge $m_{i,j}^{(l)}$ replaces $m_{p,q}^{(l)}$, and the rest of the sub-network remains fixed. Therefore, before the update we have: $m_{i,j}^{(l)} = 0$ and $m_{p,q}^{(l)} = 1$, and after the update we have: $m_{i,j}^{(l)} = 1$ and $m_{p,q}^{(l)} = 0$. We can compare the score parameters corresponding to edge $m_{i,j}^{(l)}$ and $m_{m,n}^{(l)}$ before the update as:

$$\left|s_{i,j}^{(l)}\right| < \left|s_{p,q}^{(l)}\right|, \quad \left|s_{i,j}^{(l)} + \eta \frac{\delta \log \mathcal{R}}{\delta s_{i,j}^{(l)}}\right| > \left|s_{p,q}^{(l)} + \eta \frac{\delta \log \mathcal{R}}{\delta s_{p,q}^{(l)}}\right| \tag{8}$$

where $\mathcal{R}$ can be $\mathcal{R}_P$ or $\mathcal{R}_N$ or $\mathcal{R}_C$. We have $s_{i,j}^{(l)}\frac{\delta log \mathcal{R}}{\delta s_{i,j}^{(l)}} \propto s_{i,j}^{(l)}\frac{\delta log \mathcal{R}_P}{\delta s_{i,j}^{(l)}} \propto \frac{\delta \mathcal{R}_P}{\delta P(v_j^{(l)})}P(v_i^{(l-1)})|s_{i,j}^{(l)}| \geq 0$, hence the derivative of the new edge must be higher than the old ones: $\left|\frac{\delta \log \mathcal{R}}{\delta s_{i,j}^{(l)}}\right| > \left|\frac{\delta \log \mathcal{R}}{\delta s_{p,q}^{(l)}}\right|$.

For the Path objective $\log \mathcal{R}_P$, We have:

$$\left|\frac{1}{\mathcal{R}_P}\frac{\delta \mathcal{R}_P}{\delta P(v_j^{(l)})}P(v_i^{(l-1)})\frac{|s_{i,j}^{(l)}|}{s_{i,j}^{(l)}}\right| > \left|\frac{1}{\mathcal{R}_P}\frac{\delta \mathcal{R}_P}{\delta P(v_q^{(l)})}P(v_p^{(l-1)})\frac{|s_{p,q}^{(l)}|}{s_{p,q}^{(l)}}\right| \tag{9}$$

$$\frac{\delta \mathcal{R}_P}{\delta P(v_j^{(l)})}P(v_i^{(l-1)}) > \frac{\delta \mathcal{R}_P}{\delta P(v_q^{(l)})}P(v_p^{(l-1)}) \tag{10}$$

where $P(v_i^{(l)})m_{i,j}^{(l)}\frac{\delta\mathcal{R}_P}{\delta P(v_j^{(l+1)})}$ represents the total number of effective paths containing edge $m_{i,j}^{(l)}$.

After the update, the number of effective paths is changed by: $\Delta\mathcal{R}_P = P(v_i^{(l-1)})\frac{\delta\mathcal{R}_P}{\delta P(v_j^{(l)})} - P(v_p^{(l-1)})\frac{\delta\mathcal{R}_P}{\delta P(v_q^{(l)})} > 0$. This indicates that updating parameters concerning the path objective guarantees an increase in the number of effective paths. For the Node Objective $\log\mathcal{R}_N$ or the Kernel Objective $\log\mathcal{R}_C$, we both consider a group of parameters (Node or Kernel) as N(.). In the following, we take an example of the Node Objective:

$$\left|\frac{1}{\mathcal{R}_N}\left(1-\tanh^2\left(\gamma N(v_j^{(l)})\right)\right)\gamma\frac{\delta\log\mathcal{R}_P}{\delta s_{i,j}^{(l)}}\right| > \left|\frac{1}{\mathcal{R}_N}\left(1-\tanh^2\left(\gamma N(v_q^{(l)})\right)\right)\gamma\frac{\delta\log\mathcal{R}_P}{\delta s_{p,q}^{(l)}}\right| \quad (11)$$

When $\gamma$ is sufficiently large we have:

$$\begin{cases} \left|\frac{\delta\log\mathcal{R}_P}{\delta s_{i,j}^{(l)}}\right| > \left|\frac{\delta\log\mathcal{R}_P}{\delta s_{p,q}^{(l)}}\right| & \text{if} \quad N(v_j^{(l)})=0, \quad N(v_q^{(l)})=0 \\[2mm] \left|\frac{\delta\log\mathcal{R}_P}{\delta s_{i,j}^{(l)}}\right| > 0 & \text{if} \quad N(v_j^{(l)})=0, \quad N(v_q^{(l)})>0 \\[2mm] \underbrace{0 > \left|\frac{\delta\log\mathcal{R}_P}{\delta s_{p,q}^{(l)}}\right|}_{\text{no update}} & \text{if} \quad N(v_j^{(l+1)})>0, \quad N(v_q^{(l)})=0 \\[2mm] \underbrace{0>0}_{\text{no update}} & \text{if} \quad N(v_j^{(l)})>0, \quad N(v_q^{(l)})>0 \end{cases} \quad (12)$$

As a result, the update only occurs when the node $v_j^{(l)}$ is ineffective, and after the update, it becomes effective because $\left|\frac{\delta\log\mathcal{R}P}{\delta s_{i,j}^{(l)}}\right|m_{i,j}^{(l)} > 0$ holds (since $\frac{\delta\log\mathcal{R}P}{\delta s_{p,q}^{(l)}} \geq 0$). If $v_q^{(l)}$ is ineffective before the update, the number of effective nodes increases by one, and the number of effective paths grows by $\Delta\mathcal{R}_P = P(v_i^{(l-1)})\frac{\delta\mathcal{R}_P}{\delta P(v_j^{(l)})}$. If $v_q^{(l)}$ is already effective before the update, the change in the number of effective paths is given by $\Delta\mathcal{R}_P = P(v_i^{(l-1)})\frac{\delta\mathcal{R}P}{\delta P(v_j^{(l)})} - P(v_p^{(l-1)})\frac{\delta\mathcal{R}P}{\delta P(v_q^{(l)})}$. If $m_{p,q}^{(l)}$ is the last connection at node $v_q^{(l)}$, the node becomes ineffective, leaving the number of effective nodes unchanged; otherwise, the number of effective nodes increases by one. The Kernel Objective behaves similarly, where $N(v_j^{(l)})$ is replaced with $N(m_{i,j}^{(l)})$.

When combining the Node (Kernel) Objective with the Path Objective, we obtain:

$$\left|\frac{1-\alpha+\alpha\left(1-\tanh^2\left(\gamma N(.)\right)\right)}{\mathcal{R}_N}\frac{\delta\log\mathcal{R}_P}{\delta s_{i,j}^{(l)}}\right| > \left|\frac{1-\alpha+\alpha\left(1-\tanh^2\left(\gamma N(.)\right)\right)}{\mathcal{R}_N}\frac{\delta\log\mathcal{R}_P}{\delta s_{p,q}^{(l)}}\right| \quad (13)$$

here we combine $\gamma$ with $\alpha$ as the factor for the node objective. With sufficiently large $\gamma$, we have:

$$\begin{cases} \left|\frac{\delta\log\mathcal{R}_P}{\delta s_{i,j}^{(l)}}\right| > \left|\frac{\delta\log\mathcal{R}_P}{\delta s_{p,q}^{(l)}}\right| & \text{if} \quad N(v_j^{(l)})=0, N(v_q^{(l)})=0 \quad or \quad N(v_j^{(l)})>0, N(v_q^{(l)})>0 \\[2mm] \left|\frac{\delta\log\mathcal{R}_P}{\delta s_{i,j}^{(l)}}\right| > (1-\alpha)\left|\frac{\delta\log\mathcal{R}_P}{\delta s_{p,q}^{(l)}}\right| & \text{if} \quad\quad (i) \quad N(v_j^{(l)})=0, \quad N(v_q^{(l)})>0 \\[2mm] \left|\frac{\delta\log\mathcal{R}_P}{\delta s_{i,j}^{(l)}}\right| > \frac{1}{(1-\alpha)}\left|\frac{\delta\log\mathcal{R}_P}{\delta s_{p,q}^{(l)}}\right| & \text{if} \quad\quad (ii) \quad N(v_j^{(l)})>0, \quad N(v_q^{(l)})=0 \end{cases}$$
$$(14)$$

The combined objective continues to activate ineffective nodes after the update while ensuring that the number of effective paths always increases. When $\alpha$ is high, it increases the likelihood of (i) occurring and decreases the likelihood of (ii) occurring, thus the objective focuses more on activating more effective nodes. Conversely, when $\alpha$ is low, it decreases the likelihood of (i) and increases the

---

**Algorithm 1** Differentiable Pruning at Initialization (DPaI)

---

1: **Input:** network $f(x, \mathbf{W})$, final sparsity $\rho$, iteration steps $T$, hyperparameter $\alpha, \beta, \eta$
2: Initialize the score parameters: $s_{i,j}^{(l)} \sim \mathcal{N}(0,1)$
3: Obtain layer-wise sparsity $k^{(l)}$ from $\rho$ by using the ERK method
4: **for** $t \in 1, \ldots, T$ **do**
5:      Binarize the mask on each layer: $m_{i,j}^{(l)} \leftarrow \text{Top}_{k^{(l)}}(|s_{i,j}^{(l)}|)$
6:      Compute the number of effective paths: $\mathcal{R}_P \leftarrow f(\mathbb{1}, \mathbf{M})$
7:      Compute the derivatives with respecting to each objective: $\frac{\delta \mathcal{R}_P}{\delta s_{i,j}^{(l)}}, \frac{\delta \mathcal{R}_N}{\delta s_{i,j}^{(l)}}, \frac{\delta \mathcal{R}_C}{\delta s_{i,j}^{(l)}}$
8:      Update the score parameters: $s_{i,j}^{(l)} \leftarrow s_{i,j}^{(l)} + \eta \left( (1-\alpha)\frac{\delta \mathcal{R}_P}{\delta s_{i,j}^{(l)}} + \alpha \left( (1-\beta)\frac{\delta \mathcal{R}_N}{\delta s_{i,j}^{(l)}} + \beta \frac{\delta \mathcal{R}_C}{\delta s_{i,j}^{(l)}} \right) \right)$
9: **end for**
10: **Output:** pruned network $f(x, \mathbf{M} \odot \mathbf{W})$

---

likelihood of (ii), shifting the focus towards moving connections from ineffective nodes to effective nodes, thereby promoting more effective paths. Thus, by adjusting $\alpha$, we can control the behavior of the objective to obtain the desired sub-networks.

## 3.4 Differentiable Pruning at Initialization (DPaI) Algorithm

Pruning at Initialization (PaI) methods are designed to remove weights from neural networks before training, thereby reducing training costs. Existing approaches typically involve progressively increasing pruning sparsity or employing layer-wise solutions. However, these methods often result in sub-optimal masks as the overall pruned network has never been appropriately evaluated during the pruning process. As proposed in the previous section, we introduce a novel pruning mechanism to address this challenge. This mechanism updates all masks concurrently and optimizes them directly towards a predefined target objective based on the masked parameters at a given sparsity.

**Layer-wise Sparsity Ratios.** A significant challenge in existing Pruning at Initialization (PaI) research is the distribution of overall sparsity across each network layer. Current gradient-based methods tend to allocate lower weights to layers with more parameters, leading to an unreasonable pruning ratio in these larger layers. This can result in a catastrophic phenomenon known as layer collapse, in which a whole layer is pruned, rendering all nodes and paths ineffective Frankle et al. (2021). However, recent research by Liu et al. (2022a) demonstrates that the Erdős-Rényi Kernel (ERK) method is highly efficient and effective, surpassing most contemporary iterative or dynamic approaches in assigning layer-wise sparsity ratios.

The concept of Erdős-Rényi (ER) topology was initially applied by Mocanu et al. (2018) to introduce sparsity in Multilayer Perceptron (MLP) networks. This method employs a random topology that imposes higher sparsity on larger layers. Evci et al. (2020) further extended this approach to convolutional networks, creating the ERK, which scales the sparsity of a convolutional layer proportionally to the number of neurons/channels in that layer.

In our study, we devised a method to determine the sparsity level for each convolutional layer. This is accomplished by scaling the sparsity proportionally as follows:

$$1 - \frac{n^{l-1} + n^l + w^l + h^l}{n^{l-1} n^l w^l h^l}, \tag{15}$$

where $n^l$ denotes the number of neurons or channels in the $l^{th}$ layer, while $w^l$ and $h^l$ represent the width and height of that layer, respectively. This formula is designed to adjust the sparsity in relation to the layer's structural dimensions, ensuring a balanced and effective sparsity distribution across different layers of the network.

**Differentiable Mask Updating.** In light of the objectives previously outlined, we propose a novel differentiable algorithm designed to update masks in a gradient-based manner. This approach enables direct optimization of specified metrics for targeted sparse networks at a given sparsity level. The algorithm operates as depicted in Algorithm 1. In all our experiments, we say that the algorithm has converged when i) it runs for 3000 steps or ii) the objective (e.g. the number of effective nodes, paths, and kernels does not change significantly).

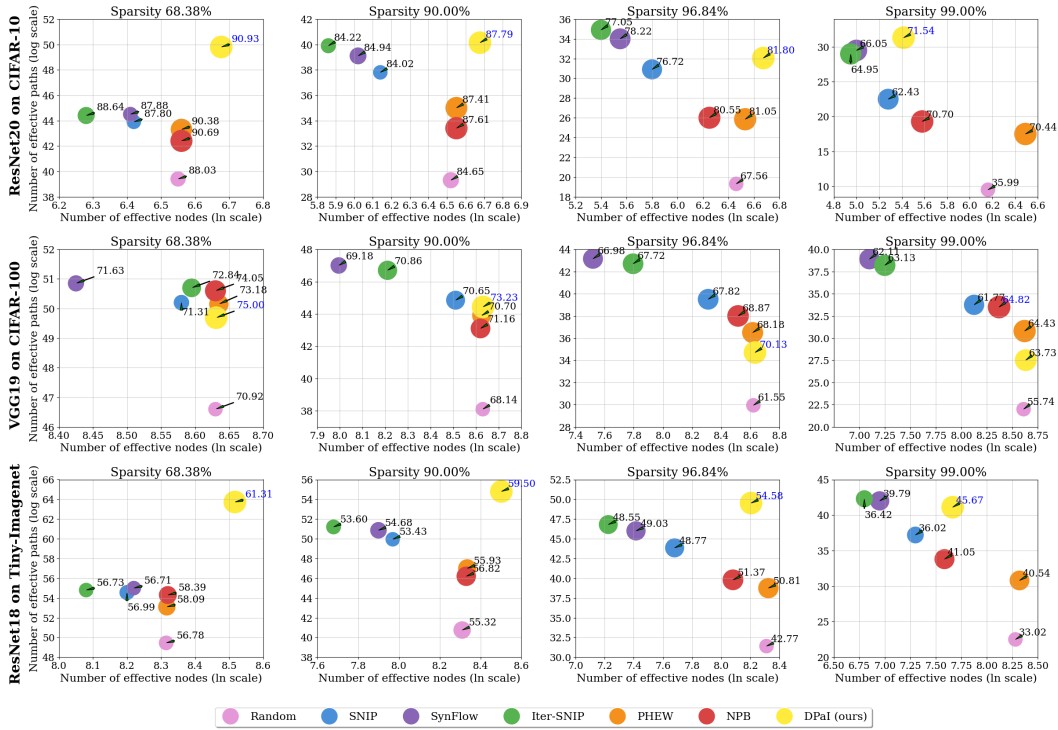

Figure 1: Accuracy of different PaI methods on three datasets with their corresponding number of effective nodes and paths across sparsity levels. The best accuracy of each setting was set in blue.

## 4 EVALUATION

In this section, we evaluate the pruned sparse network by re-training the network on different tasks (CIFAR-10, CIFAR-100, and Tiny-ImageNet, respectively) to align experiment setting present in previous state-of-the-art(SoTA) works; the training details are provided in the Appendix. C. We also perform experiments on ImageNet-1K (Deng et al., 2009) to verify our methods work on large-scale dataset tasks.

### 4.1 COMPARING DPAI WITH PREVIOUS SOTA

We now turn to evaluate the performance of DPaI. In alignment with Equation 7, we employed a grid search strategy to optimize the hyper-parameters $\alpha$ and $\beta$. This approach was instrumental in refining our final objective for the DPaI method. Subsequently, we compared DPaI with previous state-of-the-art (SoTA) methods in the Pruning at Initialization (PaI) task. Our observations reveal that DPaI consistently and significantly outperforms all prior SoTA methods across various ResNet-based tasks (see Fig. 1 for more details). This was particularly evident at higher sparsity levels (96.84%, 99.00%), where we noted the most substantial improvement in accuracy (up to 4.6%), with most cases showing improvements greater than 2%. DPaI only underperforms NPB and PHEW on the VGG19 network at 99.00% sparsity. We argue that those methods bias their algorithms towards weight magnitudes, which gives them an advantage in this specific case of high sparsity. However, DPaI, which relies solely on the Node-Path Balance principle, still outperforms all baselines at other sparsity levels on VGG19, achieving a significant gap of 1% to 2%. We can also observe how DPaI achieves substantial improvements in accuracy by comparing the number of effective nodes and effective paths identified by DPaI with those found by other baselines. In ResNet-like architectures, the subnetworks discovered by DPaI consistently have a higher number of effective nodes and paths, outperforming all baselines across all settings. Although NPB addresses the same objective using a discrete optimizer, it must decompose the full objective into layer-wise objectives due to intractability. This approach results in their method producing sub-optimal solutions for architectures with complex connections between layers, such as ResNet-like architectures with skip connections. In contrast,

Table 1: Comparison of Avg and Best Acc(%) between Synflow and DPaI Methods on ImageNet-1K

|          | Avg Acc(%)      | Best Acc(%) |
|----------|-----------------|-------------|
| Synflow  | 71.4 ± 0.29     | 71.8        |
| DPaI     | **72.2 ± 0.25** | **72.5**    |

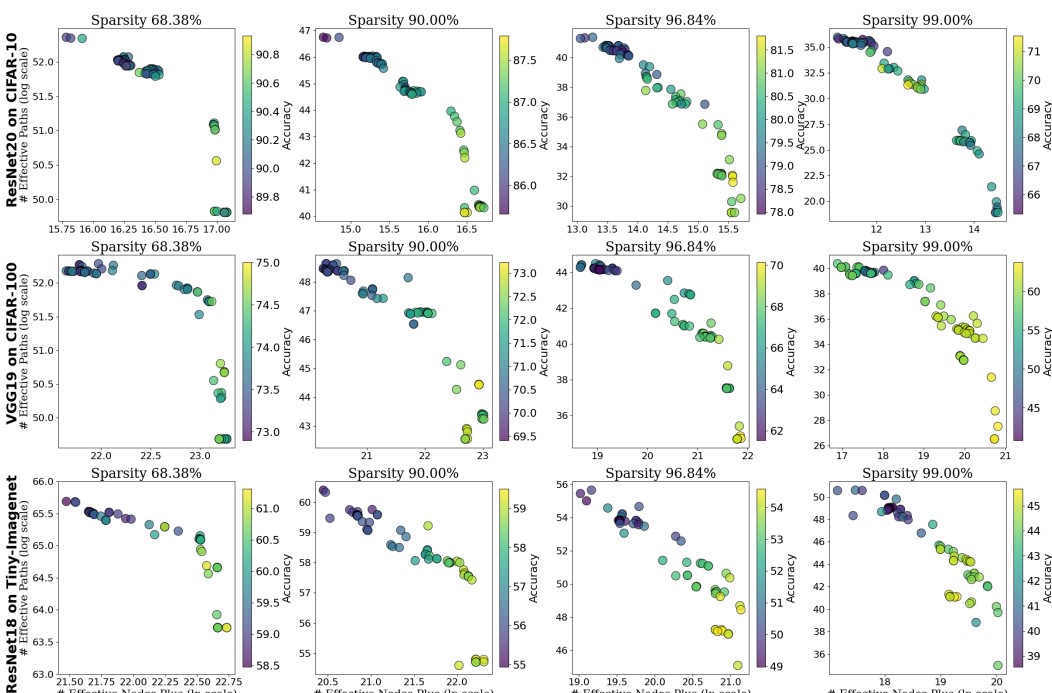

Figure 2: Results for different hyperparameter settings across various architectures and datasets, along with the number of Effective Paths (y-axis) and the combined number of Effective Nodes and Kernels (x-axis), is reported. The "# Effective Node Plus" refers to our extended concept of effective nodes, which now includes effective kernels, calculated as $\log \mathcal{R}_N + \log \mathcal{R}_C$.

DPaI can seamlessly adapt to any complex neural network architecture due to its differentiable nature. In the VGG experiments, both NPB and DPaI yield a comparable number of effective nodes and effective paths. However, simply displaying the number of effective nodes and effective paths does not fully capture the results of our objective, as it also includes the goal of activating effective kernels/connections. We present an extended view that combines nodes and kernels to analyze their impact on accuracy in Section 4.2 and Appendix H.

To verify our approach in a more practical setting, we also perform ImageNet-1K experiments on EfficientNetB0 in Table 1. We evaluated our DPaI methods against the baseline approach, Synflow, in pruning the EfficientNetB0 architecture under a sparsity constraint of 0.3. The original EfficientNetB0 model has 5.29 million parameters; our pruning target was to reduce this to 3.72 million; more experiment settings can be found in the Appendix F. Table 1 presents the average and best accuracy percentages of two methods, Synflow and DPaI, in a given experiment. The average accuracy of Synflow is 71.4% with a standard deviation of ±0.29, and its best accuracy is 71.8%. The DPaI method shows a slightly higher average accuracy of 72.2% with a standard deviation of ±0.25, and a best accuracy of 72.5%.

## 4.2 ABLATION STUDY

In Fig. 2, we analyze the effect of the hyperparameters $\alpha$ and $\beta$ on DPaI's performance. Each value of $\alpha$ and $\beta$ corresponds to a specific point in the figure, representing different numbers of effective nodes, paths, and kernels. We found that these hyperparameters highly impact DPaI's

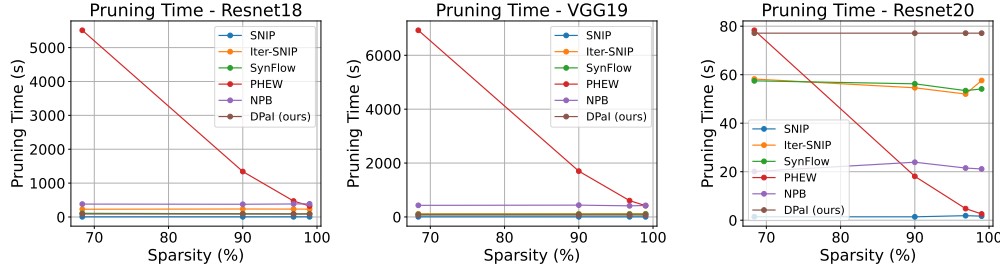

Figure 3: Pruning time of various pruning methods (SNIP, Iter-SNIP, SynFlow, PHEW, NPB, and DPaI) on different neural network architectures (Resnet18 on Tiny-ImageNet, VGG19 on CIFAR-100, and Resnet20 on CIFAR-10) across different sparsity levels.

effectiveness. However, even in the worst cases, DPaI still outperforms most baselines (such as Random, SNIP, SynFlow, Iter-SNIP) across the majority of settings. The trade-off between effective paths and effective nodes (kernels), in relation to different hyperparameter values, creates a Pareto front (Van Veldhuizen et al., 1998) in the multi-objective optimization of Node-Path-Kernel. Each point on this front represents a sub-network that is competitive with the baselines. From Fig. 2, we can identify trends that help in more quickly locating the best sub-networks on the Pareto front. The optimal sub-networks typically lie between the middle of the node-path balance point and the section with a higher number of effective nodes and kernels. A very high number of effective paths often leads to worse performance compared to having a high number of effective nodes and kernels. In conclusion, the dependency on hyperparameters is a major drawback of the PaI method, including our proposed DPaI. However, unlike most baselines (such as NPB, PHEW, SNIP, SynFlow), which bias their algorithms based on the initial weight magnitudes, DPaI is entirely data-agnostic and independent of initial weights. This makes it easier to reuse the pruned sub-network across different datasets once it has been properly pruned on a single example dataset.

### 4.3 Pruning Time of Resulting Subnetworks

We observed that DPaI has achieved significant performance gain compared to existing pruning approaches. Fig. 3 shows the pruning cost in terms of wall clock time (seconds) of the proposed DPaI as well as existing algorithms. We can observe that DPaI provided consistent and relatively low pruning time across different initial architectures and sparsity, showing our efficiency is robust across different pruning settings. Compared with previous methods like NPB, in which initial architectures significantly influence the pruning time, PHEW strongly correlated with initial architectures and the level of sparsities. For our DPaI method, we update differentiable scores using layer-specific statistics. Although implemented sequentially for simplicity, these computations can be parallelized as they are layer-independent, potentially reducing pruning time. Even without parallelization, DPaI is able to discover better subnetworks without significantly increasing pruning time. The detailed statistics of pruning time results are shown in the Appendix. B.

## 5 Conclusion

In this paper, we introduce DPaI, a novel differentiable method for pruning at initialization (PaI). This approach extended the Node-Path Balancing (NPB) principle, addressing the challenges of developing a continuous gradient for the NPB optimization problem. Extensive experimental results demonstrate that DPaI outperforms the state-of-the-art without incurring significantly higher computational costs. Due to its differentiability, DPaI's key advantage over the current best solution NPB, is its seamless integration into standard neural network training pipelines. This capability opens up potential applications for DPaI in areas like neural architecture search and sparse training. Future work will focus on customizing DPaI for these applications and exploring parallelization to further improve efficiency.

### Acknowledgments

This work was conducted while Quan Nguyen-Tri was funded by the Vingroup Innovation Foundation (VINIF) under project code VINIF.2021.ThS.BK.06.

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

| Architecture | Sparsity | | | |
|---|---|---|---|---|
| | 68.38% | 90.00% | 96.84% | 99.00% |
| ResNet20 - CIFAR-10 | 1.0/1.0 | 0.99/0.8 | 0.99/0.5 | 0.99/0.5 |
| VGG19 - CIFAR-100 | 1.0/0.1 | 0.99/0.9 | 1.0/0.5 | 1.0/0.5 |
| ResNet18 - Tiny-Imagenet | 1.0/0.1 | 1.0/0.1 | 0.99/0.5 | 1.0/0.5 |

Table 2: The best hyperparameters $\alpha/\beta$ for each experiment.

Geng Yuan, Xiaolong Ma, Wei Niu, Zhengang Li, Zhenglun Kong, Ning Liu, Yifan Gong, Zheng Zhan, Chaoyang He, Qing Jin, et al. Mest: Accurate and fast memory-economic sparse training framework on the edge. *Advances in Neural Information Processing Systems*, 34:20838–20850, 2021a.

Xin Yuan, Pedro Savarese, and Michael Maire. Growing efficient deep networks by structured continuous sparsification. In *9th International Conference on Learning Representations (ICLR)*, 2021b.

Arber Zela, Thomas Elsken, Tonmoy Saikia, Yassine Marrakchi, Thomas Brox, and Frank Hutter. Understanding and robustifying differentiable architecture search. In *International Conference on Learning Representations (ICLR)*, volume 3, pp. 7, 2020.

Hattie Zhou, Janice Lan, Rosanne Liu, and Jason Yosinski. Deconstructing lottery tickets: Zeros, signs, and the supermask. *Advances in neural information processing systems*, 32, 2019.

Michael Zhu and Suyog Gupta. To prune, or not to prune: exploring the efficacy of pruning for model compression. *arXiv preprint arXiv:1710.01878*, 2017.

## A  HYPER-PARAMETERS OF DPAI

Like other PaI methods, DPaI's hyperparameters need to be tuned for optimal performance. We observed that a sufficiently large learning rate and an adequate number of updates can help DPaI converge effectively. In all experiments, we utilized the Adam optimizer to optimize the score parameters and masks, setting the learning rate to 0.005 and the number of update steps to 3000. We performed a grid search to find the best values for $\alpha$ and $\beta$. Specifically, we searched for $\alpha$ in the set 0.1, 0.5, 0.9, 0.99, 1.0, 0.0 and $\beta$ in 0.1, 0.5, 0.8, 0.9, 1.0, 0.0. The following table 2 presents the best hyperparameters $\alpha$ and $\beta$ identified for each experiment. As seen in Figure 2, the best hyperparameters tend to prioritize the Node and Kernel Objectives.

## B  DETAIL RESULTS FOR PRUNING TIME AND NETWORK FLOPS

Table 3: Pruning time and FLOPs of subnetworks for different pruning methods and compression ratios on Resnet18 - Tiny-ImageNet.

| | Pruning time (seconds) | | | | FLOPs ($10^8$) | | | |
|---|---|---|---|---|---|---|---|---|
| Sparsity (%) | 68.38 | 90.00 | 96.84 | 99.00 | 68.38 | 90.00 | 96.84 | 99.00 |
| SNIP | 5.14 | 4.95 | 5.55 | 5.64 | 11.35 | 5.77 | 3.04 | 1.55 |
| Iter-SNIP | 229.16 | 235.34 | 233.19 | 231.23 | 10.73 | 7.05 | 3.98 | 1.97 |
| SynFlow | 108.17 | 96.18 | 91.15 | 92.60 | 14.71 | 8.91 | 4.24 | 1.50 |
| PHEW | 5511.20 | 1342.03 | 471.23 | 324.78 | 14.29 | 8.35 | 3.92 | 1.50 |
| NPB | 380.52 | 375.65 | 384.32 | 387.89 | 14.37 | 5.21 | 1.74 | 0.59 |
| **DPaI (ours)** | 88.77 | 88.77 | 88.77 | 88.77 | 14.37 | 5.20 | 1.74 | 0.60 |

For our DPaI method, we compute our loss function to update differentiable scores based on statistics from each layer. For the d-NPB component, we need to compute effective nodes and paths. Noticeably, in our implementation, we calculate those statistics sequentially over each layer for simple

Table 4: Pruning time and FLOPs of subnetworks for different pruning methods and compression ratios on VGG19 - CIFAR-100.

| | Pruning time (seconds) | | | | FLOPs ($10^7$) | | | |
|---|---|---|---|---|---|---|---|---|
| Sparsity (%) | 68.38 | 90.00 | 96.84 | 99.00 | 68.38 | 90.00 | 96.84 | 99.00 |
| SNIP | 5.15 | 4.96 | 5.12 | 4.55 | 17.952 | 7.806 | 3.686 | 1.816 |
| Iter-SNIP | 115.91 | 115.52 | 116.83 | 117.60 | 18.465 | 9.479 | 4.951 | 2.529 |
| SynFlow | 96.55 | 100.33 | 101.90 | 104.67 | 22.998 | 12.702 | 6.306 | 2.605 |
| PHEW | 6928.59 | 1699.80 | 605.65 | 417.25 | 22.108 | 11.746 | 5.611 | 2.340 |
| NPB | 430.52 | 438.20 | 412.16 | 425.33 | 22.035 | 8.773 | 2.874 | 1.046 |
| **DPaI (ours)** | 71.88 | 71.88 | 71.88 | 71.88 | 22.035 | 8.773 | 2.873 | 1.046 |

Table 5: Pruning time and FLOPs of subnetworks for different pruning methods and compression ratios on Resnet20 - CIFAR-10.

| | Pruning time (seconds) | | | | FLOPs ($10^6$) | | | |
|---|---|---|---|---|---|---|---|---|
| Sparsity (%) | 68.38 | 90.00 | 96.84 | 99.00 | 68.38 | 90.00 | 96.84 | 99.00 |
| SNIP | 1.42 | 1.40 | 1.87 | 1.68 | 17,952 | 8.323 | 3.470 | 1.709 |
| Iter-SNIP | 58.21 | 54.60 | 52.02 | 57.61 | 18,465 | 9.698 | 4.510 | 2.022 |
| SynFlow | 57.46 | 56.24 | 53.43 | 54.13 | 22,998 | 11.549 | 4.263 | 1.633 |
| PHEW | 78.31 | 18.09 | 4.78 | 2.58 | 22,108 | 10.690 | 4.110 | 1.640 |
| NPB | 20.10 | 23.91 | 21.51 | 21.13 | 22,035 | 7.642 | 2.645 | 1.122 |
| **DPaI (ours)** | 77.12 | 77.12 | 77.12 | 77.12 | 23.683 | 7.642 | 2.645 | 1.114 |

implementation, but those processes can be parallel as computations in each layer are independent. We expect a further reduction in pruning time if parallel acceleration is employed. However, despite the lack of parallel acceleration, we can already see that DPaI does not significantly increase the pruning time while obtaining significantly better subnetworks.

Besides pruning time, we find that the FLOPs reduction of subnetwork after pruning is more important than pruning before training. We have measured the FLOPs of subnetworks produced by different methods. The result indicates that our DPaI benefits from the node-path principle that can produce subnetworks with lower FLOPs as NPB than other baselines. At the same time, DPaI outperforms the existing PaI methods.

## C    EXPERIMENT DETAILS

We describe our experiment settings on architectures and datasets. We use Pytorch [1] library and conduct experiments on a single A5000.

**Datasets.** Our main experiments are conducted with CIFAR-10, CIFAR-100, and Tiny-Imagenet datasets, where:

- CIFAR-10 is augmented by normalizing per-channel, randomly flipping horizontally.
- CIFAR-100 is augmented by normalizing per-channel, randomly flipping horizontally.
- Tiny-ImageNet is augmented by normalizing per channel, cropping to 64x64, and randomly flipping horizontally.

**Architectures.** We use three different networks:

- VGG-19 is a CIFAR-100 network used in SynFlow (Tanaka et al., 2020). We choose a batch-normalization version.

---

[1] https://pytorch.org

- ResNet-20 is a 20-layer CIFAR-10 version of ResNet created by He et al. (2016). This version has added batch normalization layers before each activation function.
- ResNet-18 is a ImageNet version with 18 layers adapted from SynFlow (Tanaka et al., 2020). The first convolution has kernel size 3x3 (instead of 7x7) and max-pooling layer that follows has been removed.

We treat all of the weights from convolutional and linear layers of these networks are prunable parameters, but we do not prune the biases nor the weights in the batch normalization layers. The weights in convolutional and linear layers are initialized with Kaiming normal, while biases are initialized to be zero.

**Training details** For training on final sparse network, the hyperparameters are chosen as follows:

Table 6: Summary of the architectures, datasets, and hyperparameters used in experiments.

| Network | Dataset | Epochs | Batch | Optimizer | Momentum | LR | LR Drop, Epoch | Weight Decay |
|---|---|---|---|---|---|---|---|---|
| VGG-19 | CIFAR-100 | 160 | 128 | SGD | 0.9 | 0.1 | 10x, [60,120] | 0.0001 |
| ResNet-20 | CIFAR-10 | 160 | 128 | SGD | 0.9 | 0.1 | 10x, [60,120] | 0.0001 |
| ResNet-18 | Tiny-ImageNet | 100 | 128 | SGD | 0.9 | 0.01 | 10x, [30,60,80] | 0.0001 |

## D   LAYER-WISE EFFECTIVE NODES AND EFFECTIVE PATHS

We conducted experiments comparing ERK and Uniform layer-wise sparsity using DPaI on ResNet18 - Tiny-ImageNet, as shown in Table. 7. The results indicate that ERK consistently performs better than Uniform, particularly in extreme sparsity settings. When the network reaches 99.90% sparsity, the network with Uniform layer-wise sparsity collapses, whereas ERK still produces a trainable network. Additionally, with ERK layer-wise sparsity, the effective nodes of the learned sub-network are distributed more harmoniously—uniform layer-wise sparsity results in several bottleneck layers with a very small number of effective nodes.

Table 7: Analyzing the impact of layer-wise sparsity, employing DPaI on ResNet18 trained on the Tiny-ImageNet dataset. "Log effective paths" refers to the logarithmic scale of the number of effective paths. "Test acc" indicates test set accuracy. "Layer-wise effective nodes" represents the count of effective nodes per layer.

| Sparsity | Method | Layer-wise effective nodes | log effective paths | test acc |
|---|---|---|---|---|
| 99.00% | ERK | 64-64-64-64-64-128-128-128-128-256-256-256-256-256-512-512-512-512-512-512-200 | 62.47 | 44.93% |
|  | Uniform | 4-37-50-51-60-117-126-55-126-128-255-256-210-256-256-512-503-512-351-200 | 64.94 | 41.01% |
| 99.68% | ERK | 64-64-64-64-64-128-128-128-252-256-256-256-256-502-512-511-506-497-200 | 44.98 | 30.88% |
|  | Uniform | 6-18-22-16-21-32-66-8-56-67-144-231-63-210-217-468-503-248-482-88-200 | 66.11 | 16.33% |
| 99.90% | ERK | 41-35-50-37-59-68-114-123-77-113-140-215-238-143-215-257-385-414-285-296-197 | 49.99 | 16.33% |
|  | Uniform | 0-0-0-0-0-0-0-0-0-0-0-0-0-0-0-0-0-0-0-0-0 | - | - |

## E   EXTREMELY SPARSE NETWORKS

The extreme sparsity settings pose greater challenges due to the larger search space for the remaining parameters. Other methods often struggle to balance the number of effective nodes and effective paths. Our DPaI method consistently outperforms NPB, especially with extreme sparsity (see the results in Table 8). NPB's discrete optimization and approximate node-path balancing objectives for each layer make it difficult to find the optimal number of effective nodes and paths. In our observations, NPB could only find a maximum of 3516/1749/860 effective nodes for the 99.00%/99.68%/99.90% sparsity settings, respectively. In contrast, DPaI can discover sparse networks with higher numbers of effective nodes and effective paths, and this gap becomes more significant in higher sparsity regimes.

## F   PRUNING EFFICIENTNET ON IMAGE-NET 1K DATASET

We employed Stochastic Gradient Descent (SGD) with Nesterov momentum for the training pipeline. Each model was trained for 150 epochs with a batch size of 256. The initial learning rate was set to 0.035 and decayed by a factor of 0.99 for each epoch. The optimizer was regularized using L2

Table 8: Results from experiments conducted on ResNet18-Tiny-ImageNet under extreme sparsity settings. All experiments are conducted with 5 random seeds (0-4) and results are reported as mean and standard deviation. "effective nodes" refers to the total count of effective nodes in the network. "Log effective paths" denotes the number of effective paths on a logarithmic scale. "test acc" represents the accuracy of the test set.

| Method | Sparsity | effective nodes | log effective paths | test acc (%) |
|---|---|---|---|---|
| DPaI | 99.68% | 4974±7 | 45.12±0.24 | 30.73±0.27 |
| | 99.90% | 3496±203 | 49.96±0.93 | 15.69±0.37 |
| NPB | 99.68% | 500±23 | 76.15±0.90 | 24.33±0.19 |
| | 99.90% | 626±32 | 64.06±1.58 | 11.73±0.62 |

regularization with a coefficient of 4e-5. Each model was trained using three random seeds (0, 1, 2) to ensure robustness, and the model was trained on Nvidia A100.

## G    EXERPIMENTS ON VIT

We also add further experiments with larger architectures, such as ViT-B/16 (85 million params). We trained those networks using Tiny-ImageNet, and the results are shown in the following table:

Table 9: Results on Vision Transformer (ViT) architecture: We compare PaI with baselines such as Synflow and Random ERK on ViT-B/16 at 99% sparsity, trained from scratch on Tiny-ImageNet. We use the ViT-B/16 architecture, following the implementation in https://github.com/lucidrains/vit-pytorch, which has 85 million parameters, and train it using the SGD optimizer with a Warmup Cosine Schedule.

| Method | eff nodes | log eff paths | test acc(%) |
|---|---|---|---|
| Random ERK | 83844 | 186.54 | 17.40 |
| Synflow | 10465 | 219.27 | 29.55 |
| DPaI (ours) | 82772 | 211.91 | **35.61** |

Experiments demonstrate that our approach is effective in the linear layers of transformer architectures. However, we recognize that, in its current form, our method has not been fully explored for adaptation to self-attention layers in transformers. Adapting it to function with those layers presents a promising direction for future work.

## H    NODE-PATH BALANCING PRINCIPLE IN PAI

Here, we investigate the relationship between the number of effective paths and effective nodes (kernels) in relation to performance. When we consider only nodes ($\log \mathcal{R}_N$) or kernels ($\log \mathcal{R}_C$) with paths ($\log \mathcal{R}_P$), the results do not adhere to the node-path balancing principle, as some points with a higher number of effective paths and effective nodes (kernels) still underperform. However, when we compare performance using the new Effective Nodes Plus metric ($\log \mathcal{R}_N + \log \mathcal{R}_C$), the results demonstrate greater consistency. This suggests that incorporating effective kernels/connections as an extended definition of effective nodes into the Node-Path Balancing principle is a natural progression.

## I    METRIC DETAILS

**Effective path.**    To exactly compute the number of effective paths, we remove the batch normalization layers and initialize all the remaining parameters to $1$. Then, we put the input vector one to the network, and the number of effective paths is the sum of logits on the output layer $R = \mathbf{1}^{\top}(\prod_{\ell=1}^{L} |w_{\ell}|)\mathbf{1}$.

More precisely, we face problems with pooling layers in convolutional neural networks. With the max pooling layer, we do not modify its output. At that time, the result is the maximum number

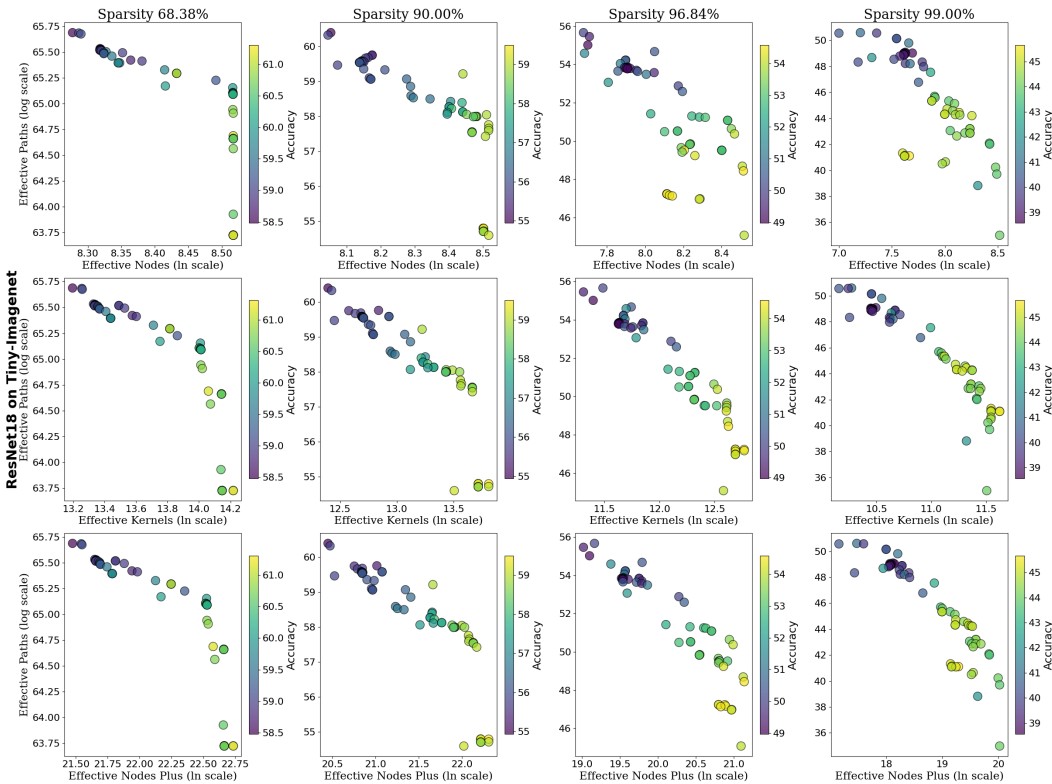

Figure 4: The accuracy for different hyperparameters settings of experiments ResNet18 on Tiny-ImageNet along with the number of Effective Paths (y-axis) and the comparison between number of Effective Nodes, Kernels and Nodes Plus (x-axis) across sparsity levels

of paths in subnetworks. With the average pooling layer, since all inputs of this layer contribute to the output, we change the average operator to the sum operator to precisely compute the number of effective paths. We all use ReLU activation functions to compute this metric since this function does not affect the calculations' results.

**Effective parameter.**   We follow Frankle et al. (2021) when identifying which is the effective parameter. Similar to computing effective paths, we make further steps. After having the sum of logits, we compute the gradients of this sum concerning weights $\nabla_{\mathbf{w}} R$. Then, if an unpruned weight has a non-zero gradient, it is effective and vice versa. Effective parameters are dense edges that connect two effective nodes

**Effective node/channel.**   With fully connected layers, if all connections to one node or out of one node are pruned, this node is pruned. If connections exist to a node, but all of these connections are ineffective, then this node becomes ineffective.

In convolutional layers, instead of nodes, we have channels. We consider a kernel as a connection and a channel as a node and then convert the convolutional layer into a fully connected layer. If all parameters in the corresponding kernel are removed, the connection is pruned. Finally, identifying the effective nodes/channels is similar to the process in fully connected layers.

## J   COMPARISON ON PREVIOUS METHODS

### J.1   COMPARING WITH SPARSITY TRAINING AND POST-TRAIN PRUNING

While the top-K operation with the Straight-Through Estimator (STE) and the Erdős-Rényi Kernel (ERK) technique are well-established and have been adopted in prior works Gao et al. (2022); Yuan

et al. (2021b); Louizos et al. (2018); Liu et al. (2022a), our contribution lies in their application and adaptation to the Pruning at Initialization (PaI) domain—a context distinct from the pruning methods used during or after training Gao et al. (2022); Yuan et al. (2021b); Louizos et al. (2018); Liu et al. (2022a); Lin et al. (2023); Veit & Belongie (2018).

Unlike traditional pruning techniques that optimize sparse masks after or during training Gao et al. (2022); Yuan et al. (2021b); Louizos et al. (2018); Liu et al. (2022a); Lin et al. (2023); Veit & Belongie (2018) using task-specific loss functions (most commonly mean squared error in image classification) with regularization terms, our proposed method, Differentiable Pruning at Initialization (DPaI), introduces differentiable masks specifically for PaI, enabling the identification of effective subnetworks before training commences. This novel application integrates network topology considerations, such as the Node-Path Balance (NPB) principle, into initialization-stage optimization. By leveraging the differentiable Node-Path Balancing (d-NPB) framework, our method balances effective nodes and paths to achieve superior trainability and performance by directly optimizing the differentiable NPB objectives and facilitating task-agnostic pruning.

Furthermore, while techniques like Gao et al. (2022); Yuan et al. (2021b); Louizos et al. (2018); Liu et al. (2022a); Lin et al. (2023); Veit & Belongie (2018) focus on pruning trained models or dynamically adjusting during training, DPaI eliminates the need for iterative pruning and retraining cycles, significantly reducing computational overhead. Regarding ERK methods, we directly adopted approaches fromLiu et al. (2022a), as they are standard techniques for NPB and PHEW; our method is designed based on those findings.

### J.2 Compared to Previous PaI Methods

Here, we briefly introduce the baseline methods used for comparison with our proposed method. The fundamental principle of Pruning at Initialization (PaI) methods is to calculate an important score for each parameter in the network and then prune the network by removing parameters with the lowest scores until the desired sparsity level is achieved.

SNIP (Lee et al., 2019a) determines importance scores by evaluating the sensitivity of each connection to the training loss. Iter-SNIP (de Jorge et al., 2021), an iterative extension of SNIP, uses the same importance scores but incrementally prunes the network from its dense state to the target sparsity level. Both SNIP and Iter-SNIP rely on data and task-specific training loss, which makes them fall under data-dependent PaI methods. SynFlow (Tanaka et al., 2020), on the other hand, is an iterative, data-agnostic PaI method designed to maintain network connectivity even at extreme sparsity levels. Similarly, PHEW (Patil & Dovrolis, 2021) is an iterative, data-agnostic PaI method that uses a random walk biased toward higher weight magnitudes to identify and preserve critical input-output paths. NPB Pham et al. (2023) introduces the node-path balancing principle, suggesting that networks with a higher number of effective nodes and paths tend to achieve better performance. They also utilise a suboptimal discrete optimization approach to identify subnetworks with a high number of effective nodes and paths.

## K Discuss Pruning Time with Model Size Increasing

The proposed DPaI method exhibits a complexity comparable to Iter-SNIP and Synflow, as all these methods iteratively update the scores of each parameter over multiple iterations. Each iteration involves a forward pass and a backward pass of the neural network. In contrast, PHEW employs a random walk strategy biassed towards higher weight magnitudes to identify input-output sets to be preserved, continuing until the subnetwork reaches the predefined sparsity. Consequently, the complexity of PHEW depends on the number of parameters in the base network and the target sparsity level. Similarly, NPB relies on a discrete optimizer, with complexity scaling based on the number of parameters in the base model. To mitigate this, NPB partitions the parameters of each layer into chunks of no more than 16,384 parameters and solves the discrete optimization for each chunk, though this partitioning often results in suboptimal solutions.

Among the architectures compared, ResNet20 has the smallest number of parameters (272,474), while ResNet18 and VGG19 have significantly larger parameter counts at 11,271,232 and 20,086,692, respectively. Consequently, DPaI, SNIP, Iter-SNIP, and Synflow demonstrate considerably lower pruning times than PHEW and NPB on ResNet18 and VGG19, underscoring their scalability for large-scale models. Moreover, since DPaI and Synflow are data-agnostic methods, they further

reduce pruning times when applied to larger-scale datasets such as Tiny-ImageNet. This highlights the superior scalability of DPaI and Synflow not only for large-scale models but also for extensive datasets.

In Figure 3, you may notice that our pruning method takes longer compared to some other methods. This is because our approach involves updating the pruning masks simultaneously in a differentiable manner during optimization. While this differentiable updating introduces additional computational overhead upfront, it allows us to maintain a relatively consistent pruning time across different model sizes.

Most previous Prune-at-Initialization (PaI) methods tend to show a significant increase in pruning time as the model size grows. This is due to their iterative or layer-wise pruning procedures, which become more time-consuming with larger models. In contrast, our method's simultaneous and differentiable mask optimization scales more efficiently with model size, ensuring that the pruning time does not increase substantially as the models become larger. As we presented in the appendix, our pruning time for different sizes of the models (ResNet18, ResNet20, VGG19) under different sparsities(from 68.38 to 99%) are in the range of (70-90)s, while NPB can increase from the 20s to 430s, and PHEW can increase from 78.31s to 6928.59s.

Therefore, although our method may take more time initially - especially compared to simpler methods on smaller models - it offers better scalability and efficiency for larger models. We believe this trade-off results in a more practical and effective pruning approach for models of varying sizes.

## L    DETAIL ABOUT OUR OBJECTIVE DESIGNING

DPaI updates the importance scores iteratively in a differentiable manner, employing STE to estimate the gradients of the importance scores in relation to the discrete optimization problem. The mechanism by which DPaI updates the importance scores to maximize the number of effective nodes and paths can be observed in the derivatives associated with each objective.

**Path Objective (Equation 2):**The derivative with respect to the importance score $s_{i,j}^{(l)}$ of a connection shows that its score increases based on the number of incoming paths $P(v_i^{(l-1)})$ and outgoing paths $\frac{\delta \mathcal{R}_P}{\delta P(v_j^{(l)})}$ it can connect. Specifically, $P(v_i^{(l-1)}) \cdot \frac{\delta \mathcal{R}_P}{\delta P(v_j^{(l)})}$ represents the number of effective paths the connection can contribute to creating. This encourages the selection of connections that can form the highest number of effective paths.

**Node Objective (Equation 6):**The derivative with respect to the node objective indicates that the importance score of a connection increases if it belongs to an ineffective node. Additionally, the score is influenced by how many effective paths the connection can contribute to creating. This dynamic promotes the selection of parameters from diverse nodes, encouraging more nodes to become effective.

These mechanisms demonstrate that the optimized importance scores effectively result in a subnetwork that maximizes the number of effective nodes and paths. Furthermore, through the convergence analysis in Section 3.3, we establish that each step of the iterative update can result in an incremental increase in the number of effective nodes and paths.

To better understand the convergence of the proposed method, we conducted empirical observations through ablation studies, examining how the number of effective nodes, paths, and kernels changes during the optimization of each objective. Table 10 shows the number of effective nodes, paths (in logarithmic scale), and kernels in the subnetwork obtained by optimizing each objective independently.

The results when pruning ResNet20 at 99.68% sparsity indicate that:

- The maximum number of effective paths (in ln scale) is **69.22**, achieved by optimizing the path objective alone. (*Visualization during training*: 5a)

- The maximum number of effective nodes is **570**, achieved by optimizing the node objective alone. (*Visualization during training*: 5b)

- The maximum number of effective kernels is **860**, achieved by optimizing the kernel objective alone. (*Visualization during training*: 5c)

| Objective | Eff. Nodes | Eff. Kernels | Eff. Paths (ln scale) |
|-----------|-----------|-----------|-----------|
| $\mathcal{R}_P$ | 75 | 211 | **69.22** |
| $\mathcal{R}_N$ | **570** | 740 | 22.30 |
| $\mathcal{R}_C$ | 168 | **860** | 47.97 |
| $\mathcal{R}_P + \mathcal{R}_N + \mathcal{R}_C$ | 321 | 822 | 42.16 |

Table 10: Number of effective nodes, kernels, and paths under different objectives.

- Optimizing the overall Node-Path Balancing (NPB) objective must strike a balance among these three objectives. (*Visualization during training*: 5d)

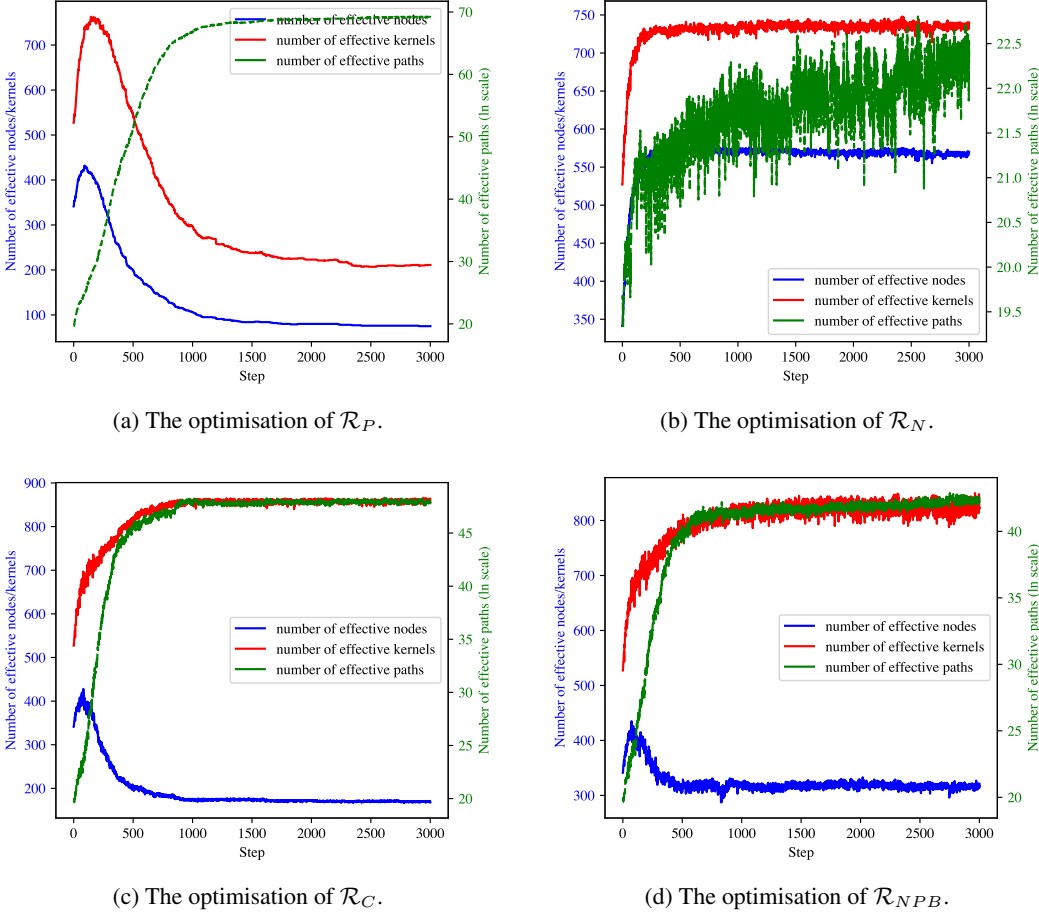

(a) The optimisation of $\mathcal{R}_P$.

(b) The optimisation of $\mathcal{R}_N$.

(c) The optimisation of $\mathcal{R}_C$.

(d) The optimisation of $\mathcal{R}_{NPB}$.

Figure 5: Visualizations of effective paths, nodes, kernels, and convergence during training.

## L.1 RATIONAL OF CHOSEN LOGARITHMIC

The main purpose of using the logarithmic scale for the number of effective nodes and effective paths in our differentiable objective is to address the imbalance between these objectives, as discussed in Section 3.2, lines 177–183. The number of effective paths is typically far greater than the number of effective nodes or connections. For example, in ResNet18 under 68.38% sparsity, the number of effective paths can reach up to $10^{65}$ (see Figure 2, ablation study on hyperparameters), while the number of nodes is only around 5000. Mathematically, the logarithmic function is an optimal choice for handling such large numbers, as $\log x$ grows the slowest as $x$ approaches infinity.

We kindly point out that the theoretical justification for employing the logarithmic scale is embedded in the computation of the derivative for each objective under the logarithmic scale. The derivative

with respect to the path objective in equation (2) is proportional by the number of effective paths that may contain $s_{i,j}^{(l)}$, which itself is a very large number similar to the total number of effective paths. However, with the logarithm scale, the derivative is now divided by the total number of effective paths, making it relatively smaller.

## L.2 RATIONAL OF CHOSEN STE

The principle of pruning revolves around identifying the importance score for each parameter and removing those with the lowest scores. Applying the Top-K function to these importance scores is a natural fit for this task, especially when pruning the network to achieve a specific sparsity level. DPaI determines the importance score in a differentiable manner by applying the Top-K function on these scores to compute the NPB objective, updating the scores using the Straight-Through Estimator (STE). While there exist soft and differentiable alternatives to the Top-K functionGao et al. (2022); Xie et al. (2020), the hard Top-K function with STE appears to be computationally more efficient and aligns well with our objective, which focuses solely on maximising the number of effective nodes and paths rather than their specific importance scores. Additionally, we provide both theoretical analysis on the convergence of the proposed objective and experimental validation, demonstrating that the method with the straightforward STE effectively optimises the number of effective nodes and paths in the pruned subnetworks.

To the best of our knowledge, DPaI is the first method to apply the Top-K function with STE to address a non-differentiable optimization problem, specifically in PaI. While previous works have utilised the Top-K function with STE for non-differentiable optimization problems in pruning during or after training, their focus has been on optimising the training loss along with a regularisation term for reducing model FLOPs. However, these approaches require extensive training of the neural network, making them unsuitable for PaI settings. In contrast, our proposed method focuses solely on optimising the network topology using the NPB objective, which is data-independent and significantly more computationally efficient.

## L.3 RATIONAL FOR TANH FUNCTION

The $\tanh(\cdot)$ function naturally aligns with our objective since it only counts the number of effective nodes. Given that $N(\cdot) \geq 0$ represents the number of effective paths passing through a node/channel or kernel/connection, the $\tanh(\cdot)$ function outputs $\tanh(N(\cdot)) = 1$ for effective nodes and $\tanh(N(\cdot)) = 0$ for ineffective nodes.

Alternatively, any activation functions $f(x)$ with similar characteristics can be employed. For instance, the sigmoid function $\sigma(x) = \frac{1}{1+e^{-x}}$ can be adapted for this purpose using $f(x) = 2\sigma(x) - 1$. This modification with sigmoid function yields derivatives computed as $2\sigma(x)(1 - \sigma(x))$, which exhibits behaviour consistent with the analysis presented in Section 3.3.

