# OpenReview forum: "DPaI: Differentiable Pruning at Initialization with Node-Path Balance Principle"
_ICLR.cc/2025/Conference — ICLR 2025 Poster_

### Official Review · Reviewer_v68w · 2024-10-29

**Soundness:** 3
**Presentation:** 3
**Contribution:** 3
**Rating:** 6
**Confidence:** 2

**Summary:**

Update:
Thanks for the authors' feedback. My concerns have been fully addressed.

----- original review -----
The authors propose a novel differentiable algorithm for deep network pruning at initialization. The key idea of the algorithm is to optimize the number of effective paths and nodes while pruning the network parameters. The non-differentiable issue can be addressed by Straight-Through Estimator. The authors demonstrate the superiority of the proposed method on several popular benchmark datasets.

**Strengths:**

The proposed algorithm presents a novel approach to Node-Path Balancing in network pruning. It distinguishes itself by incorporating this methodology into a differentiable framework. Additionally, the article provides extensive numerical results for various vision tasks and includes detailed ablation studies, offering readers valuable insights into the performance of the proposed algorithm under different conditions. Furthermore, the authors provide some convergence analysis, which can be useful for those interested in understanding the underlying principles of the method.

**Weaknesses:**

* The experiments are mostly for vision models. It would be nice to see some LLM experiments. For exmaple, how does the method perform on Llama-3.2-90B?
*  It is unclear why choosing logarithm in maximizing NPB principle. For example, why not using square root or other functions? Please suggest the theoretical support of this choice, and conduct numerical ablation studies to verify the optimality of this design. A possible approach is to make this loss function learnable, then find an optimized loss function on a set of small datasets, finally generalize to larger datasets.

* Using STE to convert non-differentiable optimization to differentiable one has been proposed in many previous works. Please justify the novelty of this approach when appling STE to this work. Any special adaptation to make STE more efficient / robust in this problem?

**Questions:**

* Why choosing logarithm in NPB principle? Is logarithm loss an optimal design in this problem?
* Is the method applicable to LLM? How it performs on popular LLMs such as Llama-3.2 families?
* Why using STE while there are several possible methods to relax the differentiable issue? For example the popular [0,1] + L1 norm relaxation seems a good choice too.

---

> ### Author Response · Authors · 2024-11-21
> **Thank you for your supportive feedback.**
>
> > Is the method applicable to LLM? How it performs on popular LLMs such as Llama-3.2 families?  … For example, how does the method perform on Llama-3.2-90B?..
>
> Thank you for raising this important question about the applicability of our DPaI method to Large Language Models (LLMs) such as the Llama-3.2 family, including models like Llama-3.2-90B.
> Our DPaI method is designed as a Prune-at-Initialization (PaI) technique, focusing on identifying the best subnetwork or substructure trained from scratch. The primary goal is to save computational resources for training while achieving performance comparable to the full model. This approach contrasts with post-training pruning methods, which aim to reduce the size of a pre-trained model by removing parameters while preserving the learned information.
> At present, we have not applied our DPaI method directly to LLMs like Llama-3.2-90B. Our experiments have focused on models and datasets where full training is computationally feasible, allowing us to evaluate and validate our approach thoroughly. Extending our method to LLMs would require significant computational resources and careful adaptation to handle the specific characteristics of those models. From the technique report released by Llama-3.2 author, training Llama-3.2-90B requires computational resources as follows:
>
> **Stage 1 pretraining: 885K H100 hours; Stage 2 annealing: 885K H100 hours; SFT: 3072 H100 hours; RLHF: 2048 H100 hours**
>
> We recognise the importance of evaluating our method on LLMs. Training LLMs is often highly costly, and PaI can potentially significantly reduce the cost of training those models (as PaI reduces model size before training). As discussed in the paper, our approach offers great efficiency concerning model size compared to other PaI methods. The differentiable mechanism we employ optimises sparse masks for the entire network simultaneously, which becomes particularly advantageous as model size increases. This suggests that our method could provide more significant benefits over existing state-of-the-art methods when applied PaI to larger models like LLMs. Our preliminary results on pruning the different sizes of the model are shown in Figure. 3 in the paper indicates that our method scales efficiently with model size, making it a promising candidate for larger models in the future.
>
> > …It is unclear why choosing logarithm in maximising NPB principle.. Why choose logarithm in NPB principle? Is logarithm loss an optimal design in this problem?
>
> The main purpose of using the logarithmic scale for the number of effective nodes and effective paths in our differentiable objective is to address the imbalance between these objectives, as discussed in Section 3.2, lines 177–183. The number of effective paths is typically far greater than the number of effective nodes or connections. For example, in ResNet18 under 68.38% sparsity, the number of effective paths can reach up to $\(10^{65}\)$ (see Figure 2, ablation study on hyperparameters), while the number of nodes is only around 5000. Mathematically, the logarithmic function is an optimal choice for handling such large numbers, as $\(\log x\)$ grows the slowest as $\(x\)$ approaches infinity.
>
>
>
> We kindly point out that the theoretical justification for employing the logarithmic scale is embedded in the computation of the derivative for each objective under the logarithmic scale. The derivative with respect to the path objective in equation (2) is proportional to the number of effective paths that may contain $s_{i,j}^{(l)}$, which itself is a very large number similar to the total number of effective paths. However, with the logarithm scale, the derivative is now divided by the total number of effective paths, making it relatively smaller.

---

> > ### Author Response · Authors · 2024-11-21
> >
> > > …Why using STE while there are several possible methods to relax the differentiable issue? For example the popular [0,1] + L1 norm relaxation seems a good choice too.
> >
> > The principle of pruning revolves around identifying the importance score for each parameter and removing those with the lowest scores. Applying the Top-K function to these importance scores is a natural fit for this task, especially when pruning the network to achieve a specific sparsity level. DPaI determines the importance score in a differentiable manner by applying the Top-K function on these scores to compute the NPB objective, updating the scores using the Straight-Through Estimator (STE). While there exist soft and differentiable alternatives to the Top-K function [1, 2], the hard Top-K function with STE appears to be computationally more efficient and aligns well with our objective, which focuses solely on maximising the number of effective nodes and paths rather than their specific importance scores. Additionally, we provide both theoretical analysis on the convergence of the proposed objective and experimental validation, demonstrating that the method with the straightforward STE effectively optimises the number of effective nodes and paths in the pruned subnetworks.
> >
> > To the best of our knowledge, DPaI is the first method to apply the Top-K function with STE to address a non-differentiable optimization problem, specifically in PaI. While previous works have utilised the Top-K function with STE for non-differentiable optimization problems in pruning during or after training, their focus has been on optimising the training loss along with a regularisation term for reducing model FLOPs. However, these approaches require extensive training of the neural network, making them unsuitable for PaI settings. In contrast, our proposed method focuses solely on optimising the network topology using the NPB objective, which is data-independent and significantly more computationally efficient.
> >
> > [1] Disentangled Differentiable Network Pruning
> >
> > [2] Differentiable Top-k Operator with Optimal Transport

---

> > > ### Author Response · Authors · 2024-11-23
> > > **Results on Language Model**
> > >
> > > > The experiments are mostly for vision models. It would be nice to see some LLM experiments. For exmaple, how does the method perform on Llama-3.2-90B? Is the method applicable to LLM? How it performs on popular LLMs such as Llama-3.2 families?
> > >
> > > We acknowledge the importance of evaluating our method on large language models (LLMs) like Llama-3.2-90B and appreciate the reviewer’s suggestion. Training LLMs is not only highly resource-intensive but also involves challenges that diverge from the core objective of PaI, which is to reduce training costs by decreasing model size prior to training. We are actively working on experiments with small-scale language models to validate the applicability of our approach in this domain.
> > >  To demonstrate the versatility of PaI beyond vision models, we conducted experiments on a smaller-scale language model. Specifically, we trained an encoder-decoder Transformer (41 million parameters) on the IWSLT14 De-En machine translation task, which contains 160K training samples. The evaluation was performed using BLEU scores under two beam search configurations (beam sizes of 1 and 5). The results are summarized in the table below:
> > >
> > > | Method | Sparsity | BLEU (beam=1) | BLEU (beam=5) |
> > > |-------------|----------|---------------|---------------|
> > >  | **Dense** | 0% | 34.49 | 35.24 |
> > >  | **SynFlow** | 90% | 29.54 | 30.64 |
> > >  | **DPaI** | 90% | **30.94** | **31.76** |
> > >
> > > Our method (DPaI) demonstrates a 1.4 BLEU improvement over SynFlow at beam=1 and a 1.12 BLEU improvement at beam=5 while maintaining performance levels close to the dense model. These results highlight the generality and effectiveness of our approach across diverse model architectures and tasks, including natural language processing. We remain committed to extending our evaluation to LLMs in the future as resources permit, further solidifying the applicability of PaI in large-scale language models.

---

### Official Review · Reviewer_gcQx · 2024-10-30

**Soundness:** 3
**Presentation:** 3
**Contribution:** 2
**Rating:** 6
**Confidence:** 5

**Summary:**

(+) The authors propose a novel pruning method referred to as Differentiable Pruning at Initialization (DPaI), which involves differentiable optimization of the pruning mask. DPaI adopts a dynamic and adaptable pruning process, allowing easier optimization processes and better solutions.

(+) DPaI’s differentiable formulation enables easy use of the gradient-based methods for Pruning at Initialization (PaI).

(+) The empirical results demonstrate that DPaI significantly outperforms current state-of-the-art PaI methods on various architectures, including Convolutional Neural  Networks (CNN) and Vision-Transformers (ViT).

**Strengths:**

(+) The author analyzes the interpretations of the proposed method (DPaI) and proves its behavior using convergence theory.

Please see the strength of the paper in summarization further.

**Weaknesses:**

(-) It isn't easy to follow how the proposed method (DPaI) finds the best nodes and paths using only an importance score.

Through extensive ablation studies on the update rule (line 8) stated in Algorithm 1, the authors should show the effectiveness: the importance of nodes and paths seems to depend only on score s_{i,j}. The empirical observations (small neural networks) with the ablation studies help us understand the contributions of the proposed method, DPaI, and the convergence theory.

(-) What about pre-trained model-based DPaI performances of ViT?

**Questions:**

Please see the above weaknesses.

---

> ### Author Response · Authors · 2024-11-21
> **Thank you for your valuable feedback.**
>
> We thank the reviewer for acknowledging the strengths of our work in methods and performance and providing more feedback for our paper's clarity.  We hope the answer below solves all the clarity issues. If you have any further questions, please sincerely welcome the reviewer involved in further discussion,
>
>
> > It isn't easy to follow how the proposed method (DPaI) finds the best nodes and paths using only an importance score.
>
> Thank you for your questions and suggestions for further clarification regarding the proposed DPaI method. DPaI updates the importance scores iteratively in a differentiable manner, employing STE to estimate the gradients of the importance scores in relation to the discrete optimization problem.  The mechanism by which DPaI updates the importance scores to maximise the number of effective nodes and paths can be observed in the derivatives associated with each objective.
>
> - **Path Objective (Equation 2):** The derivative with respect to the importance score $\( s_{i,j}^{(l)} \)$ of a connection shows that its score increases based on the number of incoming paths $\( P(v_i^{(l-1)}) \)$ and outgoing paths $\( \frac{\delta \mathcal{R}_P}{\delta P(v_j^{(l)})} \)$ it can connect. Specifically, $\( P(v_i^{(l-1)}) \cdot \frac{\delta \mathcal{R}_P}{\delta P(v_j^{(l)})} \)$ represents the number of effective paths the connection can contributes to creating. This encourages the selection of connections that can form the highest number of effective paths.
>
> - **Node Objective (Equation 6):** The derivative with respect to the node objective indicates that the importance score of a connection increases if it belongs to an ineffective node. Additionally, the score is influenced by how many effective paths the connection can contribute to creating. This dynamic promotes the selection of parameters from diverse nodes, encouraging more nodes to become effective.
> These mechanisms demonstrate that the optimised importance scores effectively result in a subnetwork that maximizes the number of effective nodes and paths. Furthermore, through the convergence analysis in Section 3.3, we establish that each step of the iterative update can result in an incremental increase in the number of effective nodes and paths.
>
> To better understand the convergence of the proposed method, we conducted empirical observations through ablation studies, examining how the number of effective nodes, paths, and kernels changes during the optimization of each objective. The following table shows the number of effective nodes, paths (in logarithmic scale), and kernels in the subnetwork obtained by optimising each objective independently.
> | Objective                          | eff nodes | eff kernels | eff paths (ln scale) |
> |------------------------------------|-----------|-------------|-----------------------|
> | $\mathcal{R}_P$                   | 75        | 211         | **69.22**             |
> | $\mathcal{R}_N$                   | **570**   | 740         | 22.30                |
> | $\mathcal{R}_C$                   | 168       | **860**     | 47.97                |
> | $\mathcal{R}_P + \mathcal{R}_N + \mathcal{R}_C$ | 321       | 822         | 42.16                |
> The results when pruning ResNet20 at 99.68% sparsity indicate that:
> - The maximum number of effective paths (in ln scale) is **69.22**, achieved by optimising the path objective alone. (visualization during training https://postimg.cc/K4QZwFkg)
> - The maximum number of effective nodes is **570**, achieved by optimising the node objective alone. (visualization during training https://postimg.cc/21tNmSrV)
> - The maximum number of effective kernels is **860**, achieved by optimising the kernel objective alone.  (visualization during training https://postimg.cc/BXXGzFfd)
> - Optimising the overall Node-Path Balancing (NPB) objective must strike a balance among these three objectives. (visualization during training https://postimg.cc/fVw6Y0T3)
>
> We have also included detailed visualisation illustrating how the number of effective nodes, paths, and kernels evolves during each training step in the revised version of the manuscript. These figures provide a clear view of the convergence behaviour of our proposed method. In summary, the results show that our objective effectively converges within approximately **3000 steps** (or even fewer in some cases). This provides empirical evidence supporting our convergence theory discussed in Section 3.3.
>
> For any confusion, the DPaI objective does not seek to find the best effective nodes and paths but rather to find the subnetwork that contains the highest number of effective nodes and paths.

---

> > ### Author Response · Authors · 2024-11-21
> >
> > > What about pre-trained model-based DPaI performances of ViT?
> >
> > Thank you for bringing up this important aspect of our work. We appreciate your suggestion to explore the performance of our DPaI method on pre-trained ViT models.
> > Typically, Prune-at-Initialization (PaI) methods have objectives that differ from those of post-training pruning methods. Post-training pruning aims to reduce the size of a pre-trained model while preserving the information contained in the trained parameters. In contrast, PaI methods focus on identifying the best subnetwork or substructure that can be trained from scratch, saving computational resources during training while achieving comparable performance.
> > As you pointed out, applying our DPaI method to pre-trained models could offer valuable insights, especially since pre-trained models like ViT provide specific initialisations for downstream tasks. We investigate this perspective in alignment with the goals of PaI, focusing on pruning pre-trained models before fine-tuning to save computational resources. To this end, we prune the pre-trained ViT-B/16 model (trained on ImageNet-1K) to 90% sparsity and subsequently fine-tune it on downstream tasks such as CIFAR-10, CIFAR-100, and Tiny-ImageNet. The pre-trained model and fine-tuning settings follow those outlined in [1]. The results of these experiments, along with detailed settings, will be included in the revised version of the manuscript.
> > | **Method** | **Dataset**       | **ViT** | **Pre-trained ViT** |
> > |------------|-------------------|---------|---------------------|
> > | Synflow    | CIFAR-10          | 77.98   | 84.09               |
> > |            | CIFAR-100         | 49.14   | 55.10               |
> > |            | Tiny-ImageNet     | 34.94   | 42.28               |
> > | DPal       | CIFAR-10          | 78.53   | 85.11               |
> > |            | CIFAR-100         | 51.33   | 58.23               |
> > |            | Tiny-ImageNet     | 36.23   | 43.31               |
> >
> > ---
> > The table compares the performance of two pruning methods, Synflow and DPal, across different configurations of Vision Transformers (ViT) on three datasets: CIFAR10, CIFAR100, and Tiny-ImageNet. Results are further divided into models trained from scratch and those pre-trained. DPal outperforms Synflow across all datasets and configurations, particularly when models are pre-trained, highlighting its robustness and effectiveness in pruning tasks. To verify the observation in the large-scale dataset, we also perform experiments on ViT-B/16 on ImageNet-1K at 90% sparsity. The following results show:
> > | **Method** | **Eff Nodes** | **Log Eff Paths** | **Test Accuracy (%)** |
> > |------------|---------------|-------------------|-----------------------|
> > | Synflow    | 47,848        | 149.80609         | 64.33                 |
> > | DPal       | 40,325        | 172.09947         | 66.27                 |
> >
> > ---
> >
> > DPaI demonstrates significantly better performance than Synflow. We hope these observations address the reviewer's concerns regarding our performance on ViT and provide a clear validation of our approach.
> >
> > [1] AN IMAGE IS WORTH 16X16 WORDS: TRANSFORMERS FOR IMAGE RECOGNITION AT SCALE

---

> > > ### Comment · Reviewer_gcQx · 2024-11-26
> > >
> > > Thanks for your rebuttals and extensive experiments. My concerns have been addressed. I will increase my score from 5 to 6.

---

### Official Review · Reviewer_CGAa · 2024-11-03

**Soundness:** 3
**Presentation:** 2
**Contribution:** 2
**Rating:** 6
**Confidence:** 2

**Summary:**

This paper extends Node-path Balancing Principle (NBP) into differentiable formulation by the learnable masks. To solve the gradient flow problem of top-k operation, authors introduce the Straight-Through Estimator (STE) technique during mask updating, while using Erdo ̋s-Rényi Kernel (ERK) technique to determine the layer-wise sparsity ratios.

**Strengths:**

1. The differentiable NPB formulation achieves competitive performance compared to previous Pruning at Initialization (PaI) methods.
2. The proposed method can be easily integrated into the previous gradient-based methods.

**Weaknesses:**

1. This paper employs methods that are well-established in prior works, such as the top-k operation with Straight Through Estimator (STE) [1, 2, 3], and the Erdős-Rényi Kernel (ERK) technique[4]. Although the differentiable masks have not been previously applied in the PaI domain, many methods[1, 2, 3, 5, 6] in pruning during/post training methods have used the differentiable masks for model compression. I suggest the authors provide a detailed discussion regarding the difference.

2. This paper uses the tanh activation function for the mask in the top-k with STE method, particularly when sigmoid and its variants are more commonly employed in previous works[1, 2, 3]. Therefore, what is the rationality of the tanh function.

3. The models and datasets employed in the paper are predominantly small-scale validations. How does the method proposed in this paper perform on larger-scale models and datasets, such as experiments using Resnet-50 on ImageNet-1k?

4. In Fig.3, why does the proposed pruning method in the paper take more time in the Resnet20 on CIFAR-10 experiments compared to other methods, while it takes less time in the Resnet18 on Tiny-ImageNet and VGG19 on CIFAR-100 experiments?

5. I suggest including a brief introduction or citation for the abbreviations of previous methods used in this paper, as relying solely on abbreviations from related work can reduce the readability of the paper for readers who are not fully familiar with PaI-related research.

[1] Disentangled Differentiable Network Pruning.
[2] Growing Efficient Deep Networks by Structured Continuous Sparsification.
[3] Learning sparse neural networks through l_0 regularization.
[4] The unreasonable effectiveness of random pruning: Return of the most naive baseline for sparse training.
[5] Filter Pruning for Efficient CNNs via Knowledge-driven Differential Filter Sampler.
[6] Convolutional Networks with Adaptive Inference Graphs.

**Questions:**

See the weaknesses.

---

> ### Author Response · Authors · 2024-11-21
> **Thank you for your constructive feedback.**
>
> We sincerely thank the reviewer for the detailed and constructive feedback. Your insights are invaluable, allowing us to refine further and clarify our work. We believe that addressing your comments will significantly enhance the quality and understanding of our paper. We hope these clarifications address your concerns. Thank you again for your thoughtful feedback, and we look forward to your further comments.
>
>
> > I suggest the authors provide a detailed discussion regarding the difference:
>
>
> We appreciate the reviewer’s observation and suggestion regarding our methods. While it is true that the top-k operation with Straight Through Estimator (STE) and the Erdős-Rényi Kernel (ERK) technique are well-established and have been adopted in prior works [1, 2, 3, 4], our contribution lies in their application and adaptation to the Pruning at Initialization (PaI) domain, a context distinct from the pruning methods used during or after training [1, 2, 3, 5, 6].
>
> Unlike traditional pruning techniques that optimise sparse masks after or during training [1, 2, 3, 5, 6] with task-specific loss function (most are MSE in image classification) with regularisation term, our proposed method (DPaI) introduces differentiable masks, specifically for PaI, allowing the identification of effective subnetworks before training begins. This novel application integrates network topology considerations, such as the Node-Path Balance (NPB) principle, into initialization-stage optimisation. By leveraging the differentiable Node-Path Balancing (d-NPB) framework, our method balances effective nodes and paths to achieve superior trainability and performance by directly optimising the differentiable-NPB objectives, which allows us to perform task-agnostic pruning. Furthermore, while techniques like [1, 2, 3, 5, 6] focus on pruning trained models or dynamically adjusting during training, DPaI eliminates the need for iterative pruning and retraining cycles, significantly reducing computational overhead. For ERK methods, we directly adopted those methods from [4], as are standard techniques for NPB and PHEW; our method is designed based on those findings.
>
> To address the reviewer’s concern, we will provide a more detailed discussion in the paper, contrasting our approach with the methods above. Specifically, we will highlight how DPaI’s use of differentiable masks and continuous optimisation uniquely benefits the PaI domain, overcoming challenges such as layer collapse, large-scale discrete optimisation, and suboptimal network topology prevalent in prior pruning approaches.
>
>
> >  what is the rationality of the tanh function.
>
>
> The $\text{tanh}(\cdot)$ function naturally aligns with our objective since it only counts the number of effective nodes. Given that $N(\cdot) \geq 0$ represents the number of effective paths passing through a node/channel or kernel/connection, the $\text{tanh}(\cdot)$ function outputs $\text{tanh}(N(\cdot)) = 1$ for effective nodes and $\text{tanh}(N(\cdot)) = 0$ for ineffective nodes.
>
> Alternatively, any activation functions $f(x)$ with similar characteristics can be employed. For instance, the sigmoid function $\sigma(x) = \frac{1}{1+e^{-x}}$ can be adapted for this purpose using $f(x) = 2\sigma(x) - 1$. This modification with sigmoid function yields derivatives computed as $f'(x) = 2\sigma(x)(1-\sigma(x))$, which exhibits behaviour consistent with the analysis presented in Section 3.3.

---

> > ### Author Response · Authors · 2024-11-21
> >
> > >  .. larger-scale models and datasets, such as experiments using Resnet-50 on ImageNet-1k?
> >
> > Thank you for suggesting experiments on larger-scale models and datasets, such as ResNet-50 on ImageNet-1k. We appreciate your interest in seeing how our method performs in these settings.
> > We would like to draw your attention to Section 4.1 and Table 1 of our manuscript, where we present experiments using EfficientNet-B0 on the ImageNet-1k dataset. These experiments are notable because most state-of-the-art Prune-at-Initialization (PaI) methods do not include full training on ImageNet-1k due to the significant computational resources required. In this context, we compared our method primarily with SynFlow, which is one of the few methods applicable at this scale.
> > We acknowledge your request for experiments involving larger-scale and widely-used models. In response to similar requests from other reviewers emphasizing the importance of pretrained model-based evaluations, we have conducted experiments to demonstrate the performance of our proposed method on larger-scale models and datasets. Specifically, we applied our method to the ViT-B/16 [1] model on the ImageNet-1K dataset at 90% sparsity. The results are as follows:
> > | Method     | Eff Nodes | Log Eff Paths | Test Acc |
> > |------------|-----------|---------------|----------|
> > | Synflow    | 47848     | 149.80609     | 64.33    |
> > | DPal       | 40325     | 172.09947     | 66.27    |
> > The proposed method, DPaI, also significantly outperforms the baseline Synflow in this large-scale setting, achieving a notable 2% performance improvement. We will include this result in the revised version of the manuscript, along with detailed descriptions of the training settings.
> >
> > [1] AN IMAGE IS WORTH 16X16 WORDS: TRANSFORMERS FOR IMAGE RECOGNITION AT SCALE
> >
> > > I suggest including a brief introduction or citation for the abbreviations of previous methods used in this paper,
> >
> >
> > Thank you for bringing this to our attention. We appreciate your suggestion to include a brief introduction or citation for the abbreviations of previous methods used in the paper and will do so in the revised version.
> >
> > Here, we provide a brief introduction to the baseline methods used for comparison with our proposed method. The fundamental principle of Pruning at Initialization (PaI) methods is to calculate an important score for each parameter in the network and then prune the network by removing parameters with the lowest scores until the desired sparsity level is achieved.
> >
> > SNIP [1] determines importance scores by evaluating the sensitivity of each connection to the training loss. Iter-SNIP [2], an iterative extension of SNIP, uses the same importance scores but incrementally prunes the network from its dense state to the target sparsity level. Both SNIP and Iter-SNIP rely on data and task-specific training loss, which makes them fall under data-dependent PaI methods. SynFlow [3], on the other hand, is an iterative, data-agnostic PaI method designed to maintain network connectivity even at extreme sparsity levels. Similarly, PHEW [4] is an iterative, data-agnostic PaI method that uses a random walk biased toward higher weight magnitudes to identify and preserve critical input-output paths. NPB [5] introduces the node-path balancing principle, suggesting that networks with a higher number of effective nodes and paths tend to achieve better performance. They also utilise a suboptimal discrete optimization approach to identify subnetworks with a high number of effective nodes and paths.
> >
> >  We will make the necessary revisions to address this in the camera-ready version of the manuscript.
> >
> > [1]  N. Lee, T. Ajanthan, and P. Torr. Snip: Single-shot network pruning based on connection sensitivity. In International Conference on Learning Representations, 2019. URL https: //openreview.net/forum?id=B1VZqjAcYX.
> >
> > [2] P. de Jorge, A. Sanyal, H. Behl, P. Torr, G. Rogez, and P. K. Dokania. Progressive skeletonization: Trimming more fat from a network at initialization. In International Conference on Learning Representations, 2021. URL https://openreview.net/forum?id=9GsFOUyUPi.
> >
> > [3] H. Tanaka, D. Kunin, D. L. Yamins, and S. Ganguli. Pruning neural networks without any data
> > by iteratively conserving synaptic flow. Advances in Neural Information Processing Systems,
> > 33:6377–6389, 2020.
> >
> > [4] S. M. Patil and C. Dovrolis. Phew: Constructing sparse networks that learn fast and generalize well without training data. In International Conference on Machine Learning, pages 8432–8442. PMLR, 2021.
> >
> > [5] Hoang Pham, The-Anh Ta, Shiwei Liu, Lichuan Xiang, Dung D. Le, Hongkai Wen, and Long Tran-Thanh. Towards data-agnostic pruning at initialization: What makes a good sparse mask? In Thirty-seventh Conference on Neural Information Processing Systems, 2023. URL https: //openreview.net/forum?id=xdOoCWCYaY.

---

> > > ### Author Response · Authors · 2024-11-21
> > >
> > > > ..In Fig.3, why does the proposed pruning method in the paper take more time..
> > >
> > > Thank you for your question regarding Figure 3 and why our proposed pruning method appears to take more time.The proposed DPaI method exhibits a complexity comparable to Iter-SNIP and Synflow, as all these methods iteratively update the scores of each parameter over multiple iterations. Each iteration involves a forward pass and a backward pass of the neural network. In contrast, PHEW employs a random walk strategy biassed towards higher weight magnitudes to identify input-output sets to be preserved, continuing until the subnetwork reaches the predefined sparsity. Consequently, the complexity of PHEW depends on the number of parameters in the base network and the target sparsity level. Similarly, NPB relies on a discrete optimizer, with complexity scaling based on the number of parameters in the base model. To mitigate this, NPB partitions the parameters of each layer into chunks of no more than 16,384 parameters and solves the discrete optimization for each chunk, though this partitioning often results in suboptimal solutions.
> > >
> > > Among the architectures compared, ResNet20 has the smallest number of parameters (272,474), while ResNet18 and VGG19 have significantly larger parameter counts at 11,271,232 and 20,086,692, respectively. Consequently, DPaI, SNIP, Iter-SNIP, and Synflow demonstrate considerably lower pruning times than PHEW and NPB on ResNet18 and VGG19, underscoring their scalability for large-scale models. Moreover, since DPaI and Synflow are data-agnostic methods, they further reduce pruning times when applied to larger-scale datasets such as Tiny-ImageNet. This highlights the superior scalability of DPaI and Synflow not only for large-scale models but also for extensive datasets.
> > >
> > > In Figure 3, you may notice that our pruning method takes longer compared to some other methods. This is because our approach involves updating the pruning masks simultaneously in a differentiable manner during optimization. While this differentiable updating introduces additional computational overhead upfront, it allows us to maintain a relatively consistent pruning time across different model sizes.
> > >
> > > Most previous Prune-at-Initialization (PaI) methods tend to show a significant increase in pruning time as the model size grows. This is due to their iterative or layer-wise pruning procedures, which become more time-consuming with larger models. In contrast, our method's simultaneous and differentiable mask optimization scales more efficiently with model size, ensuring that the pruning time does not increase substantially as the models become larger. As we presented in appendix, our pruning time for different size of model(ResNet18, ResNet20, VGG19) under different sparsities(from 68.38 to 99%) are in the range of (70-90)s, while NPB can increase from 20s to 430s, and PHEW can increase from 78.31s to 6928.59s.
> > >
> > > Therefore, although our method may take more time initially - especially compared to simpler methods on smaller models - it offers better scalability and efficiency for larger models. We believe this trade-off results in a more practical and effective pruning approach for models of varying sizes.
> > >
> > > We appreciate your interest in our work and hope this explanation clarifies the pruning time characteristics of our proposed method, as presented in Figure 3.

---

> > > > ### Comment · Reviewer_CGAa · 2024-11-26
> > > > **After rebuttal**
> > > >
> > > > Thanks for your rebuttal. Most of my concerns have been clarified and addressed. I will increase my score from 5 to 6.

---

### Official Review · Reviewer_QU2t · 2024-11-04

**Soundness:** 3
**Presentation:** 3
**Contribution:** 3
**Rating:** 6
**Confidence:** 3

**Summary:**

This paper introduces DPaI (Differentiable Pruning at Initialization), a novel method for pruning neural networks at initialization using the Node-Path Balancing (NPB) principle. Unlike traditional methods, DPaI applies a differentiable approach to optimize pruning masks, enabling efficient gradient-based optimization of sparse networks without needing training data. By balancing the number of effective nodes and paths, DPaI maintains information flow in sparse networks, achieving superior accuracy at high sparsity levels. Experimental results show that DPaI outperforms state-of-the-art methods across various network architectures and datasets, including CIFAR-10, CIFAR-100, Tiny-ImageNet, and ImageNet-1K. DPaI also achieves lower pruning time and reduced FLOPs, demonstrating its efficiency and practical applicability.

**Strengths:**

1. Differentiable Optimization: DPaI introduces a differentiable approach, making pruning more efficient and compatible with standard neural network training, unlike discrete, layer-wise pruning methods.
2. High Performance at Extreme Sparsity: By balancing effective nodes and paths, DPaI maintains high accuracy even at sparsity levels of 96-99%, outperforming traditional methods.
3. Architecture-Agnostic: DPaI works effectively across different architectures, such as ResNet, VGG, and EfficientNet, making it broadly applicable.
4. Reduced Pruning Time and FLOPs: DPaI achieves low pruning times and significantly reduces FLOPs, enhancing computational efficiency compared to other methods.

**Weaknesses:**

1. Hyperparameter Sensitivity: I assume DPaI’s performance will depend on hyperparameters which may require tuning for different datasets and sparsity levels, as this usually happens to pruning at initialization methods.
2. Implementation Complexity: DPaI involves complex gradient calculations and differentiable mask updates, which may be harder to apply to different task or models.

**Questions:**

I currently hold a positive view of this work, but I’m somewhat lacking the sense of recent progress about pruning at initialization. I believe the most critical aspect is comparing it with other works in the same track, so I may further adjust my review after considering feedback from other reviewers.

---

> ### Author Response · Authors · 2024-11-21
> **Thank you for your supportive feedback.**
>
> We thank the reviewer for the positive feedback and support of our work. We hope to have answered all of your questions satisfactorily below. Please let us know if you see any further
> issues in the paper that must be clarified or addressed.
>
> > Hyperparameter Sensitivity:
>
> Thank you for raising the concern about hyperparameter sensitivity. We acknowledge that we performed a hyperparameter search for our method, which helped us improve performance over random hyperparameter settings on average.
> To address this concern, we included a detailed ablation study on different hyperparameters in Figure 2 of our paper. The results show that results are comparable across various hyperparameter configurations and consistently outperform state-of-the-art. This suggests that our method is robust to hyperparameter choices and does not rely on specific settings to perform well.
>
> > Implementation Complexity: ..which may be harder to apply to different task or models...
>
>
> Thank you for raising the concern about implementation complexity and the applicability of our method to different tasks or models.
> Most previous Prune-at-Initialization (PaI) methods involve either iterative sparse mask selection, as seen in methods like SynFlow, or rely on large-scale discrete optimization techniques applied to each layer individually, such as in NPB. These approaches can be complex and may pose challenges when adapting to various tasks or models. Specifically, the NPB method, which relies on a discrete optimizer, faces several drawbacks related to the intractability and complexity of discrete optimization. First, it divides the overall objective into layer-wise objectives, leading to suboptimal solutions due to the need for inter-layer connectivity. Second, as the number of parameters in the base model increases, the complexity of the discrete optimisation problem grows significantly. To reduce this complexity and maintain pruning times comparable to other baselines, NPB partitions the parameters of each layer into multiple chunks, limiting each chunk to no more than 16,384 parameters, and then solves the discrete optimization for each chunk. This approach further compromises the solution quality, making the method impractical for large-scale models with billions of parameters.
> In contrast, the implementation of our proposed method is well-suited for integration into neural network training, making it easily applicable to various architectures such as ResNet, EfficientNet, and Transformer. Specifically, our method leverages the forward pass of the neural network to compute the number of effective paths and the backward pass to calculate the derivative with respect to the path objective. These forward and backward operations are efficiently implemented in widely used deep learning frameworks such as PyTorch and TensorFlow. Following this, the derivative with respect to the node objective is computed using the previously calculated path objective derivative, as outlined in Equation (6). DPaI can prune networks independently of the task data or the specific task objective function as a data-agnostic pruning-at-initialisation method.

---

### Author Response · Authors · 2024-11-21
**Transformer Results**

In response to the reviewer's request for experiments on ViT, we provide the following results:

## ViT-B/16 on ImageNet-1K: 90% Sparsity

| **Method** | **Eff Nodes** | **Log Eff Paths** | **Test Accuracy (%)** |
|------------|---------------|-------------------|-----------------------|
| Synflow    | 47,848        | 149.80609         | 64.33                 |
| DPal       | 40,325        | 172.09947         | 66.27                 |

---

## Experiments with Pre-trained ViT-B/16 Models: 90% Sparsity


| **Method** | **Dataset**       | **ViT** | **Pre-trained ViT** |
|------------|-------------------|---------|---------------------|
| Synflow    | CIFAR-10          | 77.98   | 84.09               |
|            | CIFAR-100         | 49.14   | 55.10               |
|            | Tiny-ImageNet     | 34.94   | 42.28               |
| DPal       | CIFAR-10          | 78.53   | 85.11               |
|            | CIFAR-100         | 51.33   | 58.23               |
|            | Tiny-ImageNet     | 36.23   | 43.31               |

---

## Beyond Vision Models: Experiments with  Transformer (41 million parameters) on IWSLT14 De-En (160K training samples)

Conducting experiments on LLMs is highly resource-intensive, and their evaluation often involves zero-shot in-context learning, which does not align with the goal of PaI—reducing the cost of training models. Therefore, we conducted experiments by training smaller language models from scratch to demonstrate the performance of PaI beyond vision models. Specifically, we trained an encoder-decoder transformer architecture (with 41 million parameters) on a machine translation task using the IWSLT14 De-En dataset (160K training samples). The results, evaluated using BLEU scores with beam search settings of 1 and 5 beams, are presented in the following table:

| **Method**  | **Sparsity** | **BLEU (beam=1)** | **BLEU (beam=5)** |
|-------------|--------------|-------------------|-------------------|
| **Dense**   | 0%           | 34.49            | 35.24            |
| **SynFlow** | 90%          | 29.54            | 30.64            |
| **DPaI**    | 90%          | 30.94            | 31.76            |

---
DPaI outperforms the baseline SynFlow by 1.4 BLEU at beam=1 and 1.12 BLEU at beam=5 while maintaining performance levels close to that of the dense model. This demonstrates the versatility and effectiveness of our proposed method across different model architectures and tasks.

---

### Author Response · Authors · 2024-11-25
**Expressing Gratitude to the Reviewer and Seeking Further Feedback**

Dear Reviewers,

We would like to express our sincere gratitude for your valuable feedback and constructive comments, which have significantly helped us improve our manuscript. We are pleased that you acknowledged the core novelties and strengths of our work, as listed follows:

>**Novel Differentiable Pruning Method (DPaI)**: Introducing a differentiable approach to optimize the pruning mask, making pruning more efficient and compatible with standard neural network training processes.**(Reviewer QU2t, CGAa, gcQx, v68w)**

>**High Performance at Extreme Sparsity**: Maintaining high accuracy even at extreme sparsity levels (96–99%) by balancing effective nodes and paths, outperforming traditional pruning methods.**(Reviewer QU2t, CGAa, gcQx,v68w)**

>**Architecture-Agnostic Applicability**: Effectively applying our method across various neural network architectures—including ResNet, VGG, EfficientNet, CNNs, and Vision Transformers (ViT) Language Models(LMs)—demonstrates broad utility. **(Reviewer QU2t)**

> **Reduced Pruning Time and Computational Load**: Achieving consistently lower pruning times and significantly reducing FLOPs at different initial model sizes, enhancing computational efficiency compared to other pruning techniques.**(Reviewer QU2t)**

> **Extensive Empirical Validation**: Providing comprehensive numerical results and detailed ablation studies on various vision tasks, showing significant performance improvements over state-of-the-art PaI methods.**(Reviewer QU2t, CGAa, gcQx,v68w)**

>**Theoretical Insights and Convergence Analysis**: Offering convergence theory analysis to interpret the method's behaviour, aiding in understanding the underlying principles and ensuring reliable performance.**(Reviewer gcQx,v68w)**

We also thank the reviewers for raising insightful questions and perspectives that have helped us further improve our paper. In response, we have provided additional experiments and clarifications:

>**ViT-B/16 on ImageNet-1K (90% Sparsity)**: We conducted experiments with ViT-B/16 on the ImageNet-1K dataset at 90% sparsity, demonstrating the effectiveness of our method on large-scale models and datasets.

>**Pre-trained ViT-B/16 Models (90% Sparsity)**: We evaluated our method on pre-trained ViT-B/16 models, showcasing its applicability and performance benefits when applied to pre-trained networks.

>**Language Models**: To illustrate the versatility of our approach, we conducted experiments with a Transformer model (41 million parameters) on the IWSLT14 De-En dataset (160K training samples), demonstrating the effectiveness of our method beyond vision tasks.

>To better understand the proposed method's convergence, we conducted empirical observations through ablation studies, examining how the number of effective nodes, paths, and kernels changes during the optimization of each objective.

In response to the reviewers' requests for clarity, we have also made the following clarifications and will include those texts in our paper to improve the clarity:

>**Clarified Implementation Complexity**: We discussed why our differentiable objective brings simplicity compared with previous PaI methods that require iterative mask schema and large-scale discrete optimization

>**Rationale Behind the Design of Our Optimization Objective**: We elaborated on the reasons for choosing logarithmic scaling, the Tanh function, and the use of the Straight-Through Estimator (STE) in our optimization objective, providing theoretical justification and empirical evidence.

>**Pruning Efficiency with Increasing Model Size**: We included additional analysis and discussion on how our method scales efficiently with model size, demonstrating its practicality for larger models.

>**Compared with Previous Methods**: We provided a more detailed discussion comparing our method with previous pruning methods during or after training, as well as other PaI works, highlighting the simplicity and adaptability of our approach.

Once again, we sincerely thank the reviewers for their time and effort in reviewing our paper. Your constructive feedback has been invaluable in helping us improve the quality and clarity of our work. We would like to know if you have any further concerns about our response. We would be happy to engage in any follow-up discussion and address any additional comments.

Sincerely,

The Authors

---

### Meta-Review · Area_Chair_tt5i · 2024-12-24

**Metareview:**

This paper presents Differentiable Pruning at Initialization (DPaI), a method leveraging the Node-Path Balancing principle to optimize pruning masks, achieving improvements over state-of-the-art methods across various architectures. While reviewers raised concerns about hyperparameter sensitivity, scalability to larger models, and logarithmic scaling, the authors addressed these issues through experiments, ablation studies, and clarifications. The Area Chair recommends acceptance, with suggestions to further explore DPaI's scalability to larger models and alternative functions/objectives.

**Additional Comments On Reviewer Discussion:**

The authors effectively addressed concerns about DPaI's hyperparameter sensitivity, implementation complexity, and applicability to larger models. Through additional experiments and design clarifications, they demonstrated the method's robustness, scalability, and versatility, resolving reviewers' concerns and enhancing the paper's credibility.

---

### Decision · Program_Chairs · 2025-01-22

Accept (Poster)